# From data to diagnosis: An innovative approach to epilepsy prediction with CGTNet incorporating spatio-temporal features

**Dianli Wang[1], Enping Li[2], Yang Wang[2], Zhiyang Liu[1]\*, Aixia Sun[3], Wei Wei[4], Xuning Zhang[1], Cheng Peng[1], Fengtao Wei[1]**

**1** Changchun Sci-Tech University, Changchun, China, **2** School of Electronic Information Engineering, Changchun University of Science and Technology, Changchun, China, **3** Shanghai Zhengyuan Computer Technology Co., Ltd, Shanghai, China, **4** Changchun Industrial Senior Technical School, Changchun, China

\* wangdianli20240111@126.com

## Abstract

Epilepsy affects around 50 million people globally, causing significant burdens. While many methods predict seizures, current models struggle with handling spatiotemporal features and balancing accuracy with computational efficiency.This paper introduces a novel deep learning architecture called CGTNet, which is composed of a multi-scale convolutional network, gated recurrent units (GRUs), and Sparse Transformers. It is specifically designed for analyzing elec-troencephalogram (EEG) data to predict epileptic seizures. CGTNet enhances the ability to extract spatiotemporal features from EEG signals, demonstrating its exceptional performance in seizure prediction through rigorous evaluation on the renowned CHB-MIT and SWEC-ETHZ EEG datasets. The model achieved an accuracy of 98.89%, sensitivity of 98.52%, specificity of 98.53%, an AUROC value of 0.97, and an MCC value of 0.975 on these datasets. These results not only highlight the technical innovations of CGTNet but also validate the immense potential of deep learning in processing medical signals. Our research provides an effective new tool for early detection and continuous monitoring of epilepsy, laying the foundation for advancing healthcare with artificial intelligence technology.

## 1. Introduction

Epilepsy is a chronic neurological disorder characterized by complex seizures, affecting approximately 50 million people worldwide [1] below. This disease triggers sudden, uncontrolled recurrent episodes and loss of consciousness, profoundly impacting both neonates and adults, thereby ranking it among the most common neurological disorders. The preictal period refers to the time window ranging from several minutes to tens of minutes before seizure onset, while the interictal period

**Data availability statement:** The data underlying the results presented in the study are available from https://physionet.org/content/chbmit/1.0.0/.

**Funding:** This work is supported by the Jilin Provincial Scientific and Technological Development Program, Yang Yang Foundation, Project Grant No. 20240302097GX, the Fund of Education Department of Jilin Province, Fund No. JJKH20241673KJ, and Jilin Science and Technology Development Program Project, Project No. 20230201076GX, the General Project of the Jilin Higher Education Society, Project No. JGJX2023D861, and the Jilin Vocational Education and Adult Education Teaching Reform Research Project, Project No. 2024ZCY378. The funders provided support for data collection, but had no role in data analysis, decision to publish, or manuscript preparation.

**Competing interests:** The authors have declared that no competing interests exist.

represents the interval from the postictal phase of one seizure to the beginning of the next. The primary objective of epilepsy prediction is to detect the transition phase from the interictal to the preictal state. Currently, electroencephalography (EEG) serves as the principal technical modality for epilepsy detection. This non-invasive method records the brain's electrical activity, predicting epileptic events by identifying potential changes on the scalp that reflect neuronal activity within the brain. Therefore, there is an urgent need to develop an intelligent, robust, rapid, and user-friendly epilepsy prediction method for clinical settings.

In recent years, the rapid advancement of artificial intelligence has significantly enhanced the capability for automated epileptic seizure detection and analysis using EEG data. AI-based models can predict epileptic seizures with greater accuracy and speed, not only providing critical decision support for physicians but also alleviating the physical and psychological burden on patients by enabling timely preictal interventions. Fig 1 summarizes various epilepsy detection models developed in recent years. Williamson et al. [2] employed support vector machines (SVM) to classify preictal and ictal states. However, the "black box" nature of such models poses challenges to interpretability, particularly in scenarios requiring high explanatory clarity. Building upon Williamson's approach, Zheng et al. [3] introduced the BHM method combining BEMD and Hilbert transform for epilepsy prediction. This algorithm effectively captured preictal phase synchronization changes and provided improved interpretability, though its performance may degrade when processing noisy EEG signals. Further extending this work, Watson et al. [4] developed an Epileptic Network Emulator (ENE) that utilizes phase-locked loop (PLL) networks to simulate synchronization associated with epileptic seizures. The ENE model employs approximate entropy (ApEn) for prediction, exhibiting higher sensitivity to nonlinear phase fluctuations in neural communication and demonstrating stronger noise resilience. However, its effectiveness diminishes when comparing sequences with different approximate entropy parameters.

TRUONG et al. [5] explored a generative adversarial network (GAN)-based epilepsy prediction model, incorporating short-time Fourier transform (STFT) for EEG signal preprocessing. While this unsupervised approach significantly simplified the EEG annotation process, GAN models often struggle with the diversity and complexity of epileptic seizures, resulting in suboptimal performance on novel or diverse seizure types. Choi et al. [6] developed an integrated model combining one-dimensional convolutional layers (1D CNN) and gated recurrent unit (GRU) layers for classifying preictal and interictal periods. Although this model achieved commendable epilepsy prediction accuracy through the integration of CNN and GRU, it failed to account for individual patient variations such as epilepsy type, age, gender, and brain injury, which may influence model performance. Ahmad et al. [7] implemented an autoencoder (AE) neural network model that employs encoder and decoder components to nonlinearly transform EEG signals into feature vectors, improving prediction accuracy without increasing network depth. However, the model's decoder encounters difficulties when generating features for unrepresented latent space data. Advancing in technical complexity, Ma et al. [8] proposed a multi-channel feature fusion

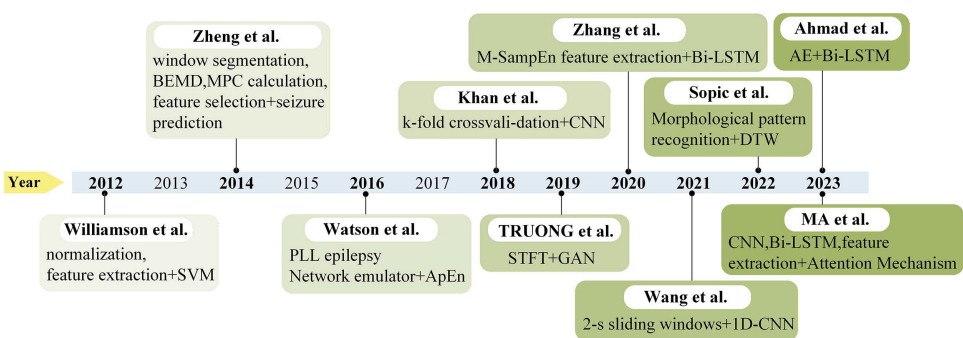

**Fig 1. Progression of modeling techniques for epilepsy prediction over the past decade.**

CNN-Bi-LSTM model that enhanced EEG signal classification accuracy, albeit at the cost of increased computational requirements and potential limitations in handling multi-scale features.

Recent developments in deep learning have further propelled advances in the field of epilepsy prediction. Esmaeilpour et al. [9] proposed a deep learning-based epilepsy prediction method that achieved high sensitivity and low false alarm rates in identifying preictal states on the CHB-MIT dataset using convolutional neural networks combined with multiple classifiers, including fully connected layers, random forests, and support vector machines. Mallick and Baths [10] introduced a novel framework combining one-dimensional convolutional layers with bidirectional LSTM and GRU, demonstrating excellent performance on the Bonn dataset with binary classification accuracy reaching 99–100% and multi-class classification accuracy ranging from 95.81% to 99.2%. Regarding feature extraction methodologies, Saadoon et al. [11] conducted a comprehensive scoping review analyzing the application of machine learning and deep learning methods in epilepsy prediction from January 2000 to April 2025, with particular emphasis on the critical role of time-domain and frequency-domain features. This review highlighted that models such as CNN, LSTM, and Transformer can significantly enhance prediction accuracy through effective extraction of time-frequency features.

Qiao et al. [12] proposed a spatiotemporal EEGNet model that integrates shrinkage spiking convolutional deep belief networks (CssCDBN) with self-attention mechanisms, employing dual-task learning to simultaneously address epilepsy prediction and detection tasks, effectively managing intra-class and inter-class variability in EEG signals. Furthermore, Kuang et al. [13] developed a feature selection method for epileptic EEG classification based on graph convolutional neural networks (GCN) and long short-term memory (LSTM) cells, which effectively extracts features from non-Euclidean spatial data by considering the topological structure among electrodes, fully leveraging information hidden within EEG signals.

Despite these advances, existing models still face challenges in balancing multi-scale feature analysis, computational efficiency, and stability, particularly when processing complex long-temporal-sequence data. This study introduces CGT-Net, which represents the first application of a combination of multi-scale CNN, GRU, and Sparse Transformer to the field of epilepsy prediction. We independently constructed two parallel convolutional architectures, each consisting of two convolutional layers with different kernel sizes, to capture both macro-level and fine-grained features across multiple scales. Additionally, CGTNet incorporates gated recurrent units (GRU), which excel at capturing long-term dependencies in sequential data, along with a Sparse Transformer component supported by a sparse self-attention mechanism that reduces computational overhead while capturing critical interrelationships within the input sequence. By introducing a sparse matrix between the positional encoder and multi-head attention mechanism, our improved Sparse Transformer mitigates over-reliance on positional information, helping prevent overfitting and enabling better learning of independent feature representations.

## 1.1. Our work

·In this study, a hybrid model named CGTNet was developed, combining multi-scale convolutional neural networks, GRU, and Sparse Transformer, aimed at improving the accuracy of epilepsy seizure prediction, particularly in handling long temporal sequences of complex EEG data.

·The model effectively captures spatial features in EEG data through a multi-scale convolutional network, while utilizing the GRU layer to process time-series data, and enhancing the ability to recognize long-term dependencies with the Sparse Transformer layer. This study introduced improvements to the Sparse Transformer by incorporating a sparse matrix between the positional encoder and the multi-head attention mechanism, allowing the model to reduce its over-reliance on positional information. This adjustment helps the model better manage noise and variations within the sequence, leading to more accurate predictions.

·This paper conducted experiments on two datasets, the CHB-MIT dataset and the SWEC-ETHZ dataset, to validate the performance of the proposed CGTNet model in the field of epilepsy prediction.

·Our experimental results indicate that this comprehensive model demonstrates superior performance in predicting epilepsy seizures compared to traditional methods, especially in identifying early or subtle signs of seizures.

## 1.2. Research objectives and paper framework

Despite significant progress in deep learning for epilepsy prediction, existing approaches still face three critical challenges: (1) insufficient multi-scale spatiotemporal feature extraction, where single-scale convolutional kernels struggle to simultaneously capture macro-level trends and micro-level details; (2) difficulty in balancing computational efficiency with accuracy, as the $O(n^2)$ complexity of traditional Transformers poses severe bottlenecks when processing long-temporal-sequence EEG data; (3) limited model generalization capability, with performance often declining significantly during cross-dataset validation. To address these issues, this study proposes the CGTNet model, which systematically tackles the challenges of feature extraction, computational efficiency, and sequence modeling through the fusion of multi-scale convolution, GRU, and Sparse Transformer.

Regarding the prediction time horizon, this study employs a 64-second window size with a 32-second sliding step to process EEG signals, with the primary objective of detecting the transition state from interictal to preictal periods. This window configuration achieves an optimal balance between model accuracy and computational complexity, effectively capturing early warning signs up to twenty minutes before seizure onset, thereby providing a critical time window for clinical intervention.The paper is organized as follows: Section 2 introduces the datasets and methodological design, including data preprocessing, multi-scale fusion strategy, GRU gating mechanism, and Sparse Transformer architecture; Section 3 presents experimental results, including performance evaluation, comparative analysis, and ablation studies; Section 4 discusses model performance and limitations; Section 5 summarizes research contributions and outlines future research directions.

## 2. Materials and methods

### 2.1. Dataset

In this research, we assessed the CGTNet model's epilepsy prediction capabilities using two prominent, publicly accessible datasets: the CHB-MIT and SWEC-ETHZ datasets.

The CHB-MIT dataset, a widely utilized EEG dataset in epilepsy research, was collaboratively compiled by the Massachusetts Institute of Technology and Boston Children's Hospital. It encompasses 23 recordings from 22 individuals (5 males, aged 3–22; 17 females, aged 1.5–19), totaling approximately 916 hours of EEG data. The EEG recordings were captured using a 10–20 electrode distribution system and documented with 18/23 leads. Specific details about the lead configuration are provided in Table 1.

**Table 1. EEG lead information.**

| Channel Label. | Channel Name. | Channel Label. | Channel Name. |
|---|---|---|---|
| Channel 1 | FP1-F7 | Channel 12 | P4-02 |
| Channel 2 | F7-T7 | Channel 13 | FP2-F8 |
| Channel 3 | T7-P7 | Channel 14 | F8-T8 |
| Channel 4 | P7-01 | Channel 15 | T8-P8 |
| Channel 5 | FP1-F3 | Channel 16 | P8-02 |
| Channel 6 | F3-C3 | Channel 17 | FZ-CZ |
| Channel 7 | C3-P3 | Channel 18 | CZ-PZ |
| Channel 8 | P3-01 | Channel 19 | P7-T7 |
| Channel 9 | FP2-F4 | Channel 20 | T7-FT9 |
| Channel 10 | F4-C4 | Channel 21 | FT9-FT10 |
| Channel 11 | C4-P4 | Channel 22 | FT10-T8 |

The CHB-MIT dataset encompasses approximately 916 hours of total EEG recordings, of which 519.6 hours represent interictal periods (non-seizure baseline states), as detailed in Table 2. The dataset was sampled with a frequency of 256 Hz and a resolution of 16 bits. It chronicles seizure episodes over varying durations, from days to months. Notably, the

**Table 2. Comprehensive details of EEG signal characteristics in the CHB-MIT dataset.**

| Patient | Channels | Interictal(h) | Seizures | mean±std(s) |
|---|---|---|---|---|
| 1 | 23 | 24 | 7 | 63±30 |
| 2 | 23 | 24 | 3 | 57±41 |
| 3 | 23 | 24 | 7 | 57±8 |
| 4 | 23 | 24 | 4 | 94±31 |
| 5 | 23 | 24 | 5 | 111±9 |
| 6 | 23 | 24 | 10 | 15±3 |
| 7 | 23 | 24 | 3 | 103±30 |
| 8 | 23 | 15 | 5 | 184±49 |
| 9 | 23 | 24 | 3 | 68±9 |
| 10 | 23 | 24 | 7 | 64±17 |
| 11 | 23 | 24 | 3 | 268±418 |
| 12 | 23 | 12 | 10 | 96±69 |
| 13 | 18 | 24 | 8 | 67±55 |
| 14 | 23 | 19 | 7 | 24±12 |
| 15 | 24 | 24 | 14 | 142±98 |
| 16 | 18 | 13 | 6 | 14±9 |
| 17 | 23 | 18 | 3 | 98±15 |
| 18 | 23 | 24 | 5 | 63±13 |
| 19 | 23 | 24 | 3 | 79±2 |
| 20 | 23 | 23.3 | 6 | 49±22 |
| 21 | 23 | 24 | 4 | 50±28 |
| 22 | 23 | 24 | 3 | 68±9 |
| 23 | 23 | 23 | 5 | 85±60 |
| 24 | 23 | 12.3 | 14 | 36±23 |
| Total | | 519.6 | 145 | |

case chb21 was recorded from the same female subject 1.5 years subsequent to case chb01. Each individual case within the dataset (such as chb01, chb02, and so forth) en-compasses 9–42 Edf files, all pertaining to a single subject. Com-prehensive details re-garding the CHB-MIT dataset are presented in Table 2. The first column on the right rep-resents the average duration of each epileptic seizure for the patient.

The CHB-MIT dataset encompasses recordings of various epileptic seizures, includ-ing focal, lateralized, and gen-eralized types. As one of the most frequently utilized epilep-sy EEG datasets, it serves as a vital resource for research and practical applications in epi-lepsy prediction, diagnosis, and treatment. Owing to its compact data size and relatively limited scope, the CHB-MIT dataset is particularly favored for model validation and comparative analysis.

Conversely, the SWEC-ETHZ dataset, amassed jointly by the Swiss Federal Institute of Technology Zurich (ETH Zurich, Switzerland) and the Swiss Epilepsy Centre (SIC), is a substantial epileptic EEG collection. It comprises 2565 hours of EEG recordings and 116 significant seizure instances from 18 patients, sampled at rates of 512 or 1024 Hz [14]. Ranking among the largest available epileptic EEG datasets, the SWEC-ETHZ dataset captures a broad spectrum of seizure types and diverse patient profiles, including varied age groups, genders, and medical histories. Its extensive data volume and comprehensive coverage render it crucial for advanced epilepsy prediction and diagnosis. Details of EEG signals from this dataset are consolidated in Table 3. The first column on the right repre-sents the average duration of each epileptic seizure for the patient.Furthermore, due to its voluminous data, multiple subgroups from the SWEC-ETHZ dataset were randomly se-lected for experimental use in this study. However, patients 12 and 20 from the CHB-MIT data-set were excluded, as their data were incompatible with the processing methods em-ployed in this research. Additionally, the data for patient 24 were deemed too limited for inclusion.

## 2.2. Methods

Fig 2 illustrates the comprehensive workflow of the CGTNet epilepsy detection method, as delineated in this study. The process encompasses three primary components: preprocessing, feature fusion, and the deep learning model

**Table 3. Detailed information of EEG signals in SWEC-ETHZ EEG dataset.**

| Patient | Channels | Interictal(h) | Seizures | mean±std(s) |
|---------|----------|---------------|----------|-------------|
| 1 | 88 | 24 | 2 | 601±17 |
| 2 | 66 | 24 | 2 | 88±2 |
| 3 | 64 | 24 | 4 | 64±4 |
| 4 | 32 | 24 | 14 | 41±14 |
| 5 | 128 | 24 | 4 | 16±1 |
| 6 | 32 | 24 | 8 | 45±33 |
| 7 | 75 | 24 | 4 | 69±38 |
| 8 | 61 | 24 | 7 | 219±176 |
| 9 | 48 | 24 | 17 | 67±47 |
| 10 | 32 | 24 | 16 | 75±21 |
| 11 | 32 | 24 | 2 | 91±11 |
| 12 | 56 | 24 | 9 | 146±33 |
| 13 | 64 | 24 | 7 | 102±61 |
| 14 | 24 | 24 | 16 | 96±39 |
| 15 | 98 | 24 | 2 | 94±35 |
| 16 | 34 | 24 | 5 | 190±51 |
| 17 | 60 | 24 | 2 | 97±1 |
| 18 | 42 | 24 | 5 | 199±100 |
| Total | | 432 | 126 | |

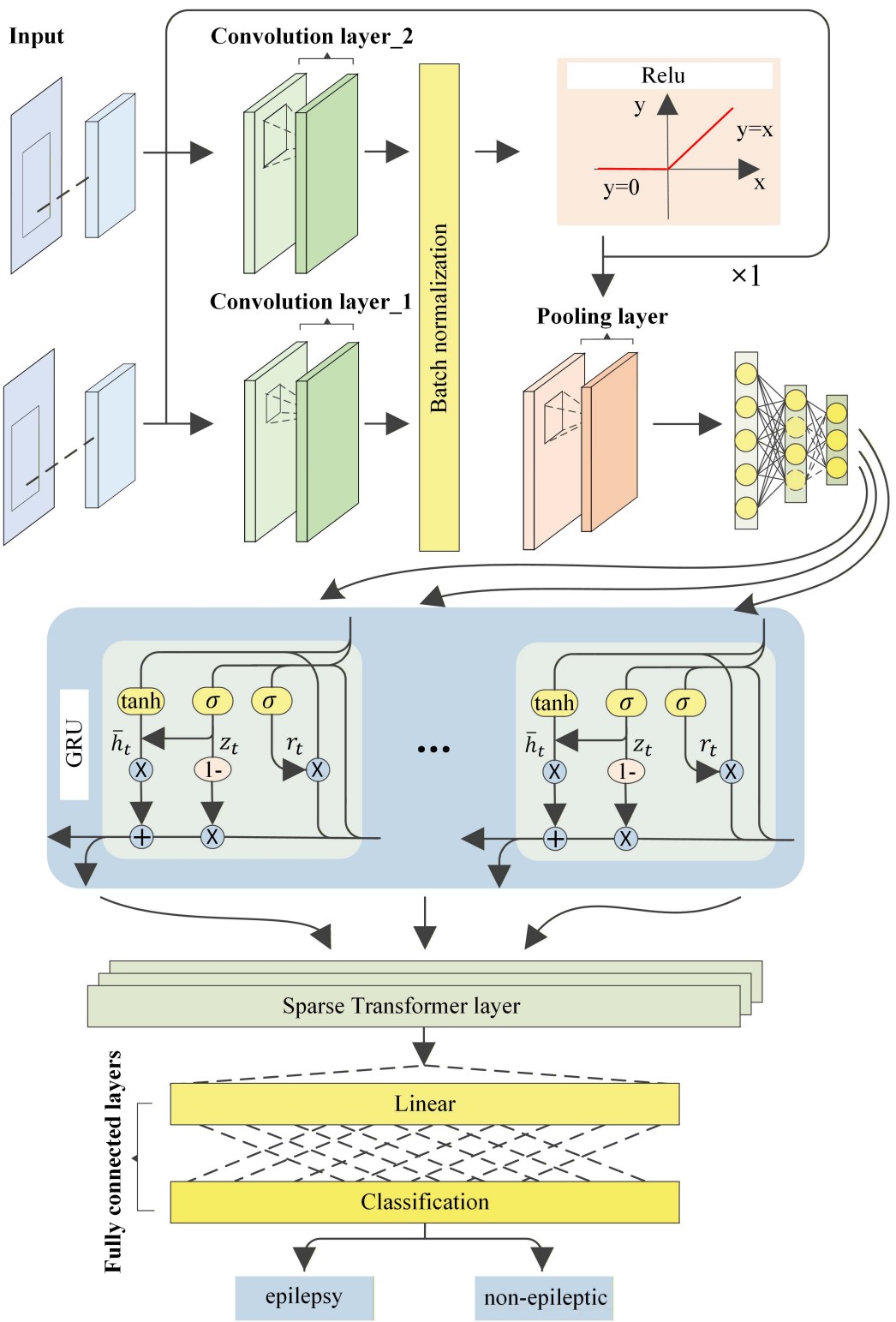

**Fig 2. Workflow diagram of the CGTNet model.**

application.Our primary evaluation unit is at the patient level. In our experiments, we independently performed 5-fold cross-validation for each patient, ensuring that each patient's data appeared only in their own training set or test set, but never in both simultaneously. This patient-specific validation strategy prevents data leakage and accurately assesses the model's generalization capability on unseen patient data. Specifically, for the CHB-MIT dataset, independent modeling and validation were performed separately for 21 patients, while for the SWEC-ETHZ dataset, independent validation was conducted on 6 randomly selected patients. Multi-channel EEG recordings in this research were segmented into 64-second intervals, These data segments are first preprocessed using the Short-Time Fourier Transform (STFT) to preserve the temporal and frequency domain information of the EEG signals and to highlight the dynamic changes in the signal frequency characteristics. The ensuing stage involves dimensionality reduction through factor analysis, facilitating the extraction of pivotal spatio-temporal features vital for epilepsy prediction. Ultimately, these refined features are input into the CGTNet model, which, through its intricate deep neural network architecture, excels in the accurate recognition and classification of epileptic seizure patterns.

To provide a more detailed and intuitive understanding of the technical depth of this paper, the following pseudocode is provided for readers to better comprehend and implement. In our research, the configuration and hyperparameter settings for CGTNet are as follows: the random seed is set to 8 to ensure reproducibility of results; a 5-fold cross-validation method is used for robust validation; the batch size for each training session is set to 64 to balance computational efficiency and model performance; training is conducted for 150 epochs to ensure thorough training and model convergence; the network size is set to 32 to accommodate the complexity of EEG data and feature extraction needs; the Adam optimizer is used with a learning rate of 0.0001 to ensure stable and effective model training. The Training process is shown in Table 4.

In the design of our model, we integrate multi-scale convolutional kernels to comprehensively capture the diverse features involved in epilepsy seizure prediction. Larger kernels (such as 30 and 32) are employed to capture macro-level characteristics within EEG signals, while smaller 3x3 kernels provide a finer level of feature extraction. Through experimental validation, using an input dimension of 2176 and a GRU with 128 hidden units, combined with a 2-layer, 4-head sparse self-attention mechanism within a Sparse Transformer structure, we achieved an optimal balance between model performance and computational efficiency. This design effectively addresses the complex spatiotemporal relationships inherent in epilepsy prediction and demonstrated outstanding performance in our experiments.

## 2.3. Data preprocessing

In this study, we chose a 64-second window size, considering the characteristics of EEG signals and the fact that epileptic seizure precursors typically occur within a time range of several minutes to tens of minutes. As depicted in Fig 3, a window size of 64 seconds was employed, indicating that each data segment spanned a 64-second duration. A 64-second window effectively captures sufficient temporal context information while avoiding the drawbacks of overly long windows that dilute features, or overly short windows that lose context. By comparing different window sizes (such as 32 seconds, 64 seconds, and 128 seconds), we found that the 64-second window strikes a good balance between model accuracy, sensitivity, specificity, and computational complexity.

Given the high dimensionality of EEG signals after preprocessing, we applied factor analysis for linear dimensionality reduction to retain the primary information related to epileptic seizures. Through experimentation to optimize the number of factors, we found that 64 factors achieved a good balance between preserving the main features of the signal and reducing noise, avoiding the issue of insufficient features when the factor number is too low or introducing redundancy when it is too high, which could affect the model's generalization ability.

Additionally, to improve the model's robustness and generalization capability, we combined traditional signal processing methods with data-driven techniques. We applied Short-Time Fourier Transform (STFT) and filtering to the EEG signals, preserving the time-frequency characteristics of the signals while reducing noise interference. This approach enhanced the model's performance across different datasets.

**Table 4. EEG data training process.**

| Algorithm: EEG Data Preprocessing, Training Process, and Network Architecture |
| --- |
| **Input:** |
| EEG data $X \in \mathbb{R}^{n \times c \times h \times w}$ |
| Training set $D = \{(x^{(n)}, y^{(n)})\}_{n=1}^{N}$, validation set $v$ |
| Learning rate $\alpha$, regularization coefficient $\lambda$ |
| Number of network layers L, number of neurons per layer $M_l$, $1 \le l \le L$ |
| function PREPROCESS_RESAMPLE: |
| # Resample all EEG data to 256 Hz |
| For each file with sampling rate ≠ 256 Hz: |
| Apply bandpass filter (1–100 Hz) |
| Resample to 256 Hz using scipy.signal.resample |
| Return resampled_data |
| end function |
| function PREPROCESS_SPECTROGRAM_GENERATION: |
| For each file, apply bandstop and highpass filters, generate spectrograms |
| Save spectrograms to disk |
| end function |
| function TRAIN_MODEL: |
| Initialize model parameters and structure, including GRU and Sparse Transformer layers |
| For training over dataset D: |
| Execute forward and backward propagation for each batch |
| Update model parameters |
| Evaluate model performance on validation set $V$ |
| end function |
| function PARALLELCONVOLUTION_NETWORK($x$): |
| $x_1 = ReLU(BN(Conv1D(x, W_{11}, stride = 1, padding = 0)))$ |
| $x_1 = ReLU(BN(Conv1D(x, W_{12}, stride = 1, padding = 1)))$ |
| $x_2 = ReLU(BN(Conv1D(x, W_{21}, stride = 1, padding = 0)))$ |
| $x_2 = ReLU(BN(Conv1D(x, W_{22}, stride = 1, padding = 1)))$ |
| $x = Concatenate(x_1, x_2, axis = 2)$ |
| $x = GRU(x, h_0, parameters = \theta\_GRU)$ |
| $x = sparseTransformerEncoder(x, parameters = \theta\_Transformer)$ |
| $x = Linear(x, W\_out, b\_out)$ |
| $\hat{y} = Softmax(x)$ |
| Return $\hat{y}$ |
| end function |
| Main Procedure: |
| Call PREPROCESS_SPECTROGRAM_GENERATION() |
| model = TRAIN_MODEL() |
| predictions = model.Predict() |
| end Main Procedure |
| Output: $Y_{classify}(\omega) \in \mathbb{R}^{n \times c' \times 2}$ represents the dimension of the output classes |

In the pre-processing stage of the EEG signal, we took a multi-step filtering process to improve data quality and reliability. First, we applied a band-stop filter to remove frequencies from 120 to 125 Hz and 60–65 Hz, which helped remove possible power line interference or other high-frequency noise. We then used a high-pass filter (with a cutoff frequency of 1 Hz) to remove low-frequency noise. These filtering steps ensure that we can extract a clear and reliable signal from the EEG data.

In the preprocessing stage, to facilitate the extraction of spatiotemporal features from EEG data, we subsequently applied the STFT algorithm for further preprocessing of clear EEG data. The essence of STFT is the Fourier transform with windowing, assuming that the non-stationary EEG signal is stationary within the short interval of the analysis window. By moving the window function along the time axis, the EEG signal is analyzed in segments to obtain a set of local spectra. STFT approximates non-stationary signals as stationary by computing the Fourier transform within a window.

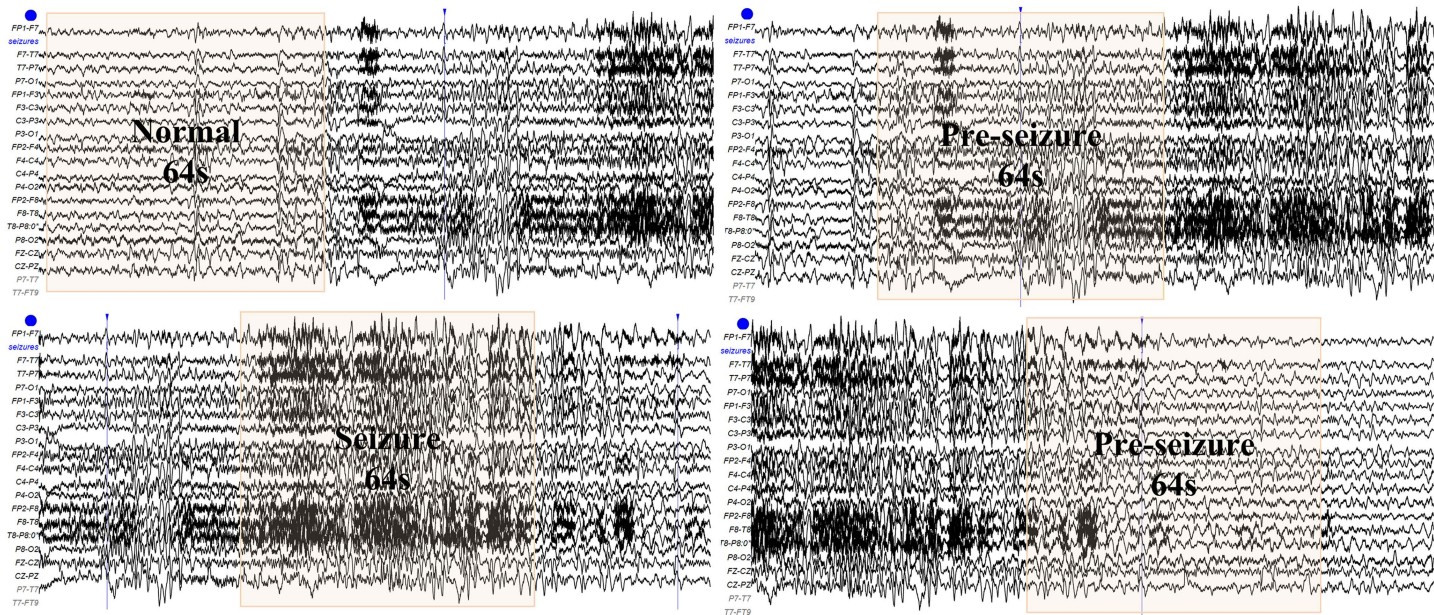

**Fig 3. CHB-MIT waveforms, illustrating the process of capturing EEG signals through a win-dow of 64 seconds size moving in steps of 32 seconds using the STFT algorithm.**

Considering the large duration of time-series data in the two datasets used in this paper, CHB-MIT (containing approximately 916 hours) and SWEC-ETHZ (containing 2565 hours), using a 5-second window size would lead to excessively long data processing and model training times. The five-fold cross-validation accuracy of seizure prediction for each patient was mostly above 97%, as shown in Tables 5 and 6. This validates that EEG data frequency domain analysis can be effectively performed using a 64-second window size and a 32-second step size for STFT processing. This parameter

**Table 5. Experimental hardware and software environment.**

|  | Attributes |
|---|---|
| Processor | Intel(R) Core(TM) i9-10920X CPU@3.50GHz |
| Disc | 1T SSD |
| RAM | 128GB |
| Operation system | Windows 11 |
| Development Languages | Python 3.9 |
| Deep learning frameworks | Pytorch 1.10 |
| GPU | NVIDIA GeForce RTX3090 |

**Table 6. K-fold cross-validation table.**

| CHB-MIT | | | | | SWEC-ETHZ | | | | |
|---|---|---|---|---|---|---|---|---|---|
| K-fold | ACC | AUROC | Sen | F1-Score | K-fold | ACC | AUROC | Sen | F1-Score |
| K = 5 | 98.71 | 0.97 | 98.65 | 99.05 | K = 5 | 99.06 | 0.98 | 98.38 | 99.09 |
| K = 10 | 97.23 | 0.96 | 99.34 | 98.05 | K = 10 | 98.26 | 0.98 | 99.45 | 98.08 |
| K = 15 | 98.28 | 0.98 | 97.43 | 98.10 | K = 15 | 99.34 | 0.97 | 97.47 | 98.16 |

choice ensures that the EEG data within each window remains stationary. The definition of the signal's $x(t)$ Short-Time Fourier Transform is (1):

$$STFT(t, f) = \int_{-\infty}^{\infty} x(\tau)h(\tau - t)e^{-j2\pi f\tau} d\tau$$

(1)

In the formula, $h(\tau - t)$ represents the analysis window function. In our experiment, we set the sampling rate to 256 Hz and the window size to 64. This parameter configuration is able to retain the temporal and frequency domain information of the EEG data comprehensively while allowing for fast computation.

In order to analyse the data in depth and extract key features, we used factor analysis. Time-variant, multi-channel EEG data possess redundant information and high complexity [15], making direct feature extraction from large datasets slow and computationally intensive for deep learning networks. To facilitate signal processing, it is essential to perform spatiotemporal decomposition of EEG data in the preprocessing stage. Therefore, the factor analysis method we employed can decompose multi-channel EEG signals into distinct components. Factor analysis refines variables with certain correlations into fewer common factors, specifically those factors with eigenvalues greater than 1, which are strongly related to epilepsy seizures [16]. Representing the original variables with these common factors not only extracts the main features but also reduces data dimensionality, decreasing the computational load on the CGTNet network for epilepsy prediction and significantly enhancing model computational efficiency. Unlike traditional dimensionality reduction algorithms, this method does not result in extensive data loss, thus preserving the validity of the results. Furthermore, in this experiment, we processed EEG data from 22 channels. However, the spatial and temporal features of some brain channels are not significantly correlated with epilepsy prediction. Statistical analysis of the correlation between each common factor and epilepsy seizures allows us to eliminate irrelevant data, such as non-specific brain activity caused by electrode motion, eye movements, respiratory rhythms, and heart rate variability, thereby removing redundant information from EEG data.

To ensure consistency and reproducibility in epilepsy prediction, we implemented a systematized decision framework. The output layer of CGTNet employs a softmax activation function to generate probability distributions for interictal and preictal states, with a binary classification threshold of 0.5 applied to the softmax output. Specifically, when the predicted preictal probability P(pre-ictal) ≥ 0.5, the sample is classified as preictal; otherwise, it is classified as interictal. This threshold was optimally selected based on the validation set to balance sensitivity and specificity.

Following the detection of a preictal state and subsequent seizure occurrence, a 30-minute refractory period is implemented to prevent redundant warnings for the same epileptic event and align with clinical observations that seizures typically do not occur in rapid succession. To further enhance clinical applicability, we implemented a post-processing pipeline that filters out predictions occurring within 5 minutes before seizure onset (insufficient intervention time) and validates predictions made 20 minutes before seizure onset as successful early warnings, providing adequate time for clinical interventions such as administering rescue medication or ensuring patient safety.

By combining sliding window techniques, fine-grained filter processing, STFT algorithm, and factor analysis, we were able to efficiently extract useful episodic and intermittent data from long-term EEG recordings. This provides a rich, high-quality data sample for subsequent model training and testing, thus helping to improve the accuracy and efficiency of epilepsy detection and prediction.

In this study, to ensure data consistency and the comparability of the research results, we decided to exclude data from patients 12, 20, and 24 in the CHB-MIT dataset. Specifically, the records for patients 12 and 20 contained formatting issues, such as the presence of a 25-hour time span, which was inconsistent with other standard records. As a result, their data were excluded. Patient 24's data was relatively sparse, which may have limited its contribution to model training, so it was also excluded. While these exclusions may affect the model's adaptability to some extent, this approach helps maintain the consistency of the study and enhances the model's generalization ability. It is worth noting that similar

exclusion strategies have been applied in many studies in related fields to ensure data quality and the stability of research outcomes.

## 2.4. Multi-scale information fusion strategy

This section is dedicated to detailing the application of a multi-scale information fusion strategy in epilepsy prediction modelling. This strategy is based on the core idea that EEG signals contain complex information at multiple frequencies and time scales, and their characterisation at different scales is crucial for accurate prediction of epileptic seizures.

In our model, The first part is a parallel multi-scale convolution module. This structure is adopted to process input data in parallel, increasing the network's width and reducing model training time. The upper and lower convolution layers use kernel sizes of 32 and 62, respectively. This design aims to extract features from the input data at different scales, capturing both macro and micro-level characteristics. This provides diverse and compact feature representations for the subsequent structures in the model. The size of the receptive field can be calculated using the following equation (2) [17]:

$$F = ((n-1) \times s) + k \tag{2}$$

where $n$ represents the number of layers in the convolutional layer, $s$ is the step size, and $k$ is the size of the convolutional kernel. This calculation allows us to determine the sensory field of each convolutional block and thus accurately tune the model to capture features at different scales.

The strategy is implemented by two parallel convolutional blocks, each targeting a different information scale for feature extraction. The mathematical representation of the strategy can be summarised as (3) [18]:

$$X_{fusion} = Concat(Conv_{large}(X), Conv_{small}(X)) \tag{3}$$

where $Conv_{large}(X)$ and $Conv_{small}(X)$ represent the outputs of the large-scale and small-scale convolutional blocks, respectively, and $X_{fusion}$ denotes the fused multiscale features.

This fusion approach is predicated on the notion that by amalgamating information across various scales, the model attains a more comprehensive understanding and interpretation of EEG signals. This aspect is particularly vital for detecting subtle physiological changes indicative of impending seizures. Compared to conventional single-scale analysis methods, this multi-scale fusion strategy markedly enhances the model's capability to decipher complex EEG data. As a result, it substantially bolsters the accuracy in predicting seizures.

## 2.5. Gate mechanism and GRU

The gating mechanism in the GRU unit consists of two dynamic gates (Fig 4): the reset gate $R_t$ and the update gate $Z_t$. The computation of these gates involves the input data $x_t$, the hidden state of the previous time step $h_{(t-1)}$, and the corresponding weight matrices and bias terms. Specifically, the matrix expression of the gating mechanism is as follows (4)~(5) [19,20]:

$$R_t = \sigma(W_r \cdot x_t + U_r \cdot h_{(t-1)} + b_r) \tag{4}$$

$$Z_t = \sigma(W_z \cdot x_t + U_z \cdot h_{(t-1)} + b_z) \tag{5}$$

where $W_r$, $W_z$, $U_r$, $U_z$ are weight matrices, $b_r$, $b_z$ are bias terms, and $\sigma$ represents the sigmoid activation function.

The new hidden state $h_t$ is computed by combining the previous hidden state $h_{(t-1)}$ and the current input $x_t$ as expressed below (6):

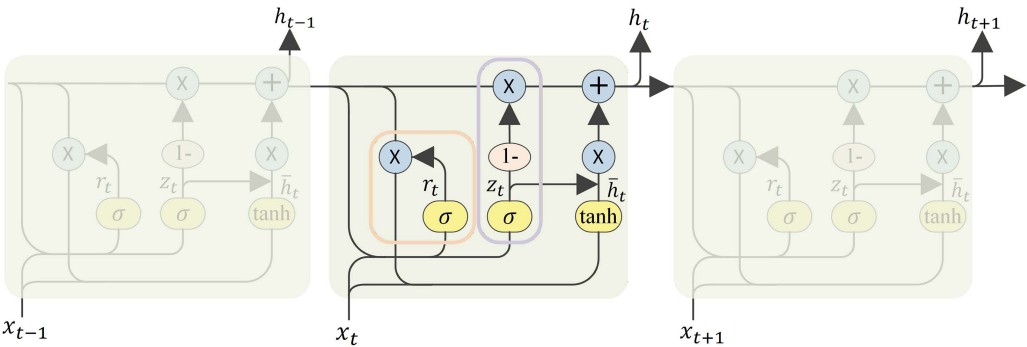

**Fig 4. Structure of GRU gate control unit.**

$$h_t = (1 - Z_t) \odot h_{(t-1)} + Z_t \odot \tanh(W_h \cdot x_t + R_t \odot (U_h \cdot h_{(t-1)} + b_h)) \tag{6}$$

where $W_r$, $W_z$, $U_r$, $U_z$ are weight matrices, $b_r$, $b_z$ are bias terms, and $\sigma$ represents the sigmoid activation function.

The new hidden state $h_t$ is computed by combining the previous hidden state $h_{(t-1)}$ and the current input $x_t$ as expressed below (7):

$$h_t = (1 - Z_t) \odot h_{(t-1)} + Z_t \odot \tanh(W_h \cdot x_t + R_t \odot (U_h \cdot h_{(t-1)} + b_h)) \tag{7}$$

Here, $W_h$, $U_h$ is the weight matrix required for the hidden state update, $b_h$ is the corresponding bias term, and $\odot$ denotes the Hadamard product (i.e., element-level multiplication).

The GRU's gating mechanism adeptly manages the equilibrium between retaining and updating information, thereby circumventing the issue of gradient vanishing and preserving long-term dependencies. This characteristic renders the GRU exceptionally well-suited for handling datasets characterized by intricate time dependencies, like EEG signals. In our CGTNet model, integrating the GRU layer has notably enhanced the capacity to detect temporal patterns associated with seizures.

In this study, we retained positional embeddings, even though the STFT-processed signals were converted into spectrogram form, resulting in some loss of original time series information. The purpose of retaining positional embeddings is to recover part of the sequential order information in the time series after GRU processing. By incorporating positional embeddings, we ensure that the subsequent sparse self-attention mechanism can utilize sequence information to model global dependencies.

Moreover, the output vector dimension of the GRU layer is set to 128, which matches the input dimension of the Sparse Transformer encoder, ensuring dimensional consistency across layers. This issue of dimensional matching within the data flow has been rigorously validated and will be detailed in this section. Regarding the choice of attention heads, we used 4 attention heads with a sparse structure, with the dimension of each head determined according to the dimension of the GRU output vector.

## 2.6. Hierarchical structure of sparse transformer encoder

In our study, the hierarchical architecture of the Sparse Transformer encoder plays a crucial role in processing complex EEG data. By leveraging its sparse self-attention mechanism and structured hierarchy, the Sparse Transformer encoder effectively captures long-term dependencies in EEG signals. Traditional Transformers use a dense attention mechanism with a computational complexity of $O(n^2)$, where $n$ represents the length of the input sequence. However, this approach

encounters significant computational and memory bottlenecks when processing high-dimensional EEG data and long sequences. To address this issue, the Sparse Transformer introduces sparse matrices to reduce computational complexity, lowering the attention calculation complexity from $O\left(n^2\right)$ to $O\left(n \cdot k\right)$, where $k$ is the number of non-zero elements. By designing specific sparse distribution patterns based on prior knowledge of the sequence data, the Sparse Transformer becomes more flexible in adapting to sequence data, thereby improving computational efficiency. This architecture is intuitively detailed in Fig 5.

In the sparse attention mechanism, the attention scores are calculated using the formula (8):

$$\text{Attention}\left(Q, K, V\right) = \text{softmax}\left(\frac{QK^T}{\sqrt{d_k}} \odot M\right) V \tag{8}$$

Here, the matrix $M$ is the sparse mask matrix, defined as follows in (9):

$$M_{ij} = \begin{cases} 1, & \text{if } (i, j) \text{ is selected} \\ 0, & \text{otherwise} \end{cases} \tag{9}$$

The sparse mask matrix $M$ determines which elements participate in the attention calculation. By sparsifying the matrix, the model focuses only on the most meaningful information for sequence prediction, thus reducing computational costs. This mechanism not only improves computational efficiency but also prevents unnecessary information transfer, mitigating the risk of model overfitting and ensuring stability when handling complex spatiotemporal data.

We introduce an information-theoretic constraint in this mechanism, mathematically represented as follows in (10):

$$I\left(X; Z\right) \leq \sum_{i=1}^{k} I\left(X_i; Z_i\right) \leq k \cdot C \tag{10}$$

Here, $X$ is the input sequence, $Z$ is the attention output, $k$ is the number of non-zero connections, and $C$ is the upper bound on the information capacity of a single connection. This constraint limits information transfer in sparse attention, ensuring computational efficiency without sacrificing essential information.

Multi-head sparse attention further enhances the model's ability to capture information from multiple perspectives. Its mathematical representation is as shown in (11):

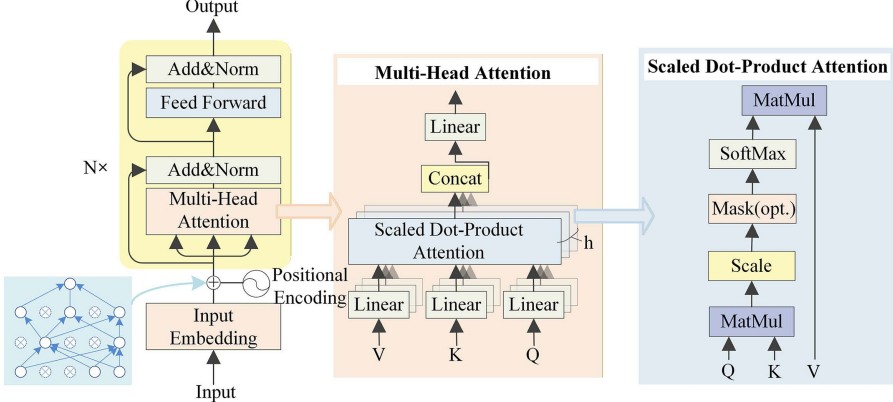

**Fig 5. Sparse Transformer encoder structure.**

$$\text{MultiHead}(Q, K, V) = \text{Concat}(\text{head}_1, \ldots, \text{head}_h) W^O \tag{11}$$

Each head's sparse attention computation is defined as:

$$\text{head}_i = \text{Attention}\left(QW_i^Q, KW_i^K, VW_i^V\right) \odot S_i \tag{12}$$

where $S_i$ is the sparse pattern matrix for the $i$-th head, used to further filter the attention results. Through this multi-head sparse mechanism, the model can capture critical information from different perspectives across multiple layers in parallel, enabling more flexible and efficient information extraction.

By introducing the sparse attention mechanism, the Sparse Transformer encoder achieves an optimal balance between computational efficiency and information capture ability. This design significantly reduces computational costs while maintaining sensitivity to long-term dependencies and subtle changes in complex EEG data. As a result, our epilepsy prediction model demonstrates enhanced adaptability and generalization capabilities, maintaining both high efficiency and accuracy when processing complex spatiotemporal data.

### 2.7. Hardware and software environments

The hardware and software environments used for the experiments in this paper are shown in Table 5.

### 2.8. Evaluation criteria

This section may be divided by subheadings. It should provide a concise and precise description of the experimental results, their interpretation, as well as the experimental conclusions that can be drawn.

In this study, in order to comprehensively measure the performance of the developed model, we introduced a comprehensive set of evaluation metrics, including Accuracy, AUROC, Sensitivity and Specificity. The formulae for these indicators are as follows:

1. Accuracy:

   - Expressed as $Acc = \frac{TP+TN}{TP+TN+FP+FN}$, which is used to measure the proportion of samples correctly classified by the model.

2. Sensitivity:

   - Expressed as $Sen = \frac{TP}{TP+FN}$, reflecting the ability of the model to identify actual positive samples.

3. Specificity:

   - Expressed as $Spe = \frac{TN}{TN+FP}$, denotes the ability of the model to identify actual negative samples.

4. F1 Score:

   - Expressed as: $F1 = \frac{2TP}{2TP+FP+FN}$, which provides a balanced evaluation metric, particularly useful for imbalanced datasets.

5. MCC:

   - The formula is $MCC = \frac{(TP \times TN)-(FP \times FN)}{\sqrt{(TP+FP) \times (TP+FN) \times (TN+FP) \times (TN+FN)}}$, which measures the classification performance of the model while also taking into account the issue of imbalanced datasets.

In these equations, "True Positives" (TP) refer to the count of samples accurately identified as positive, while "True Negatives" (TN) denote the number of samples correctly classified as negative. "False Positives" (FP) represent the

instances where samples are incorrectly labeled as positive, and "False Negatives" (FN) are the cases where samples are erroneously categorized as negative. Utilizing these metrics provides an extensive overview of the model's performance, covering crucial areas like its recognition capacity and false positive rate.

To ensure the robustness and statistical significance of our results, we employed appropriate statistical tests for model comparison. For AUROC comparison between CGTNet and baseline methods, we used DeLong's test, which is specifically designed for comparing the area under two or more correlated receiver operating characteristic curves. For comparing F1-scores and MCC values across different models, we applied the Wilcoxon signed-rank test, a non-parametric statistical hypothesis test suitable for paired samples. Statistical significance was set at $p < 0.05$. All statistical analyses were performed using Python's scipy.stats and scikit-learn libraries.

## 3. Results

### 3.1. Evaluation prediction performance

To mitigate potential overfitting or underfitting in multimodal models, a range of regularization techniques and hyperparameter tuning methods were applied in the experiments. In the CNN module, Batch Normalization was employed to accelerate training and improve the model's generalization ability. The choice of learning rate also reflects an understanding of each module's characteristics. For the CNN component, an initial learning rate of 0.0001 was selected, which helps stabilize the convergence of local features and prevents underfitting caused by excessively fast convergence. After pooling and dimensionality reduction through fully connected layers, the CNN module works in conjunction with global features from the GRU and Transformer modules for joint modeling. In the Sparse Transformer module, special attention was paid to optimizing the number of attention heads and encoder layers. A 2-layer encoder with 4 attention heads was chosen, striking a balance between performance and computational complexity, with the goal of preventing overfitting of global features.

During our research, we opted for 150 training epochs, a decision based on a comprehensive observation of the training process and multiple experimental results. The training loss decreased rapidly within the first 60 epochs and then stabilized, while the validation loss exhibited a gradual decline, indicating that the model continued to optimize during extended training. The longer training period allowed the model to fully learn deep features and optimize performance, thereby enhancing generalization capability. The loss curves are illustrated in Fig 6. We employed nn.BCEWithLogitsLoss() as the loss function, which combines the Sigmoid activation function with Binary Cross Entropy, making it well-suited for binary classification problems, particularly when handling class imbalance. During backpropagation, we computed gradients and updated weights for each layer of the CNN, GRU, and Sparse Transformer to ensure effective optimization across all parts of the model. To prevent overfitting, we utilized Dropout as a regularization technique. The experimental results demonstrated that continuing the training up to 150 epochs led to further performance improvements.

K-fold cross-validation is a commonly used model validation technique in the field of machine learning. In this study, we employed the K-fold validation method to test the effectiveness of the model. It was observed that the choice of the K value directly impacts the model's generalization ability and stability. Across the two datasets, the performance with K = 5 was relatively better, though the results with K = 10 and K = 15 also demonstrated the model's robustness, particularly when dealing with more complex validation sets. As shown in Table 6, K = 5 may be an ideal choice, especially when the sample size is limited.

Our study demonstrates the notable diagnostic accuracy of the CGTNet model when applied to the CHB-MIT epilepsy prediction dataset. As detailed in Table 7, a five-fold cross-validation method was utilized to thoroughly assess various patient samples. The model consistently exhibited an average accuracy of 98.71%, along with achieving 98.65% sensitivity and 97.9% specificity. Patients 12 and 20 from the CHB-MIT dataset were excluded from this study because they were not compatible with the processing methods used. Additionally, patient 24 was excluded due to insufficient data.

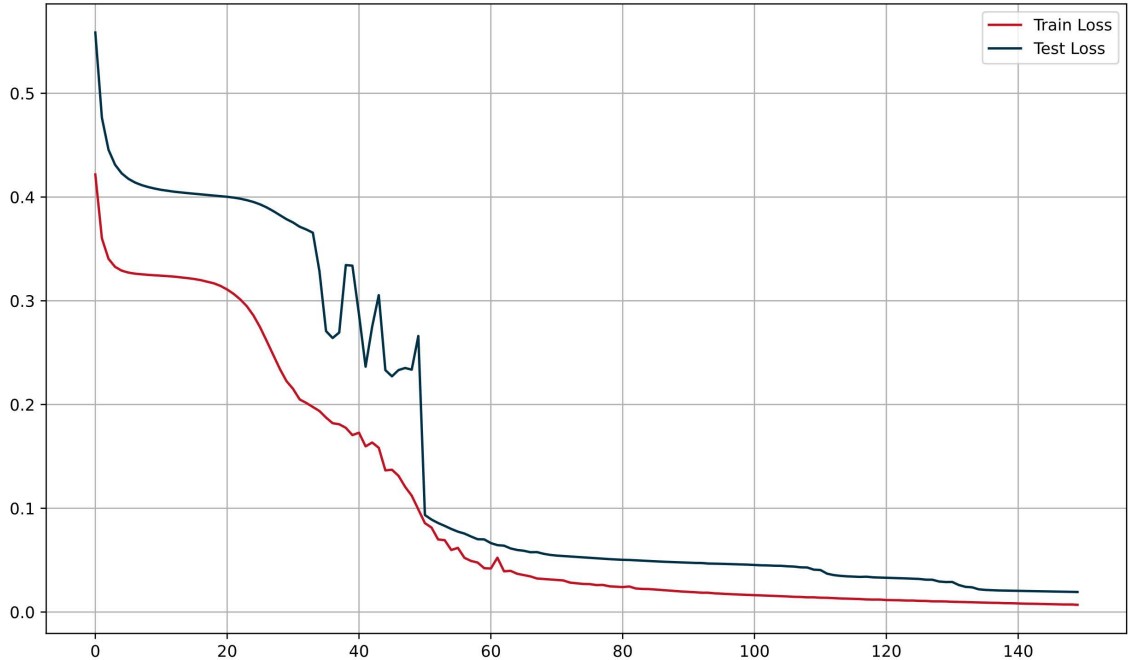

**Fig 6. CGTNet model loss curves.**

Particularly noteworthy are the results from patients 19 and 23, where CGTNet attained 99.9% accuracy and 100% sensitivity and specificity, underscoring its exceptional reliability in epilepsy detection.

The model's efficiency was further substantiated by detailed case analyses. For instance, in the case of patient 4, a 22-year-old, the model showed remarkable adaptability to complex developmental EEG patterns, indicated by an AUROC value of 0.997. Similarly, for patient 3, aged 14, who may present EEG characteristics unique to adolescence, the model maintained high accuracy with an AUROC value of 0.996. These findings affirm CGTNet's proficiency in accurately identifying seizure patterns across different developmental stages.

The CGTNet model is adept at managing the intricacies related to gender differences in epilepsy detection. For instance, in the case of a 1.5-year-old female (patient6), the model proficiently identified seizure signals, notwithstanding her early developmental stage. This showcases the model's resilience in accommodating variations in age and gender. Likewise, with a 3.5-year-old male (patient8), CGTNet exhibited a keen sensitivity to early developmental physiological characteristics. These instances highlight CGTNet's reliability in processing EEG data across diverse gender and age groups, underlining its versatility in epilepsy prediction.

In subsequent experiments, we focused on evaluating CGTNet's applicability to a different dataset, namely SWEC-ETHZ. To this end, we analyzed 400 hours of EEG recordings to confirm the model's sustained high performance across diverse datasets. Several patients were randomly selected from the SWEC-ETHZ dataset for five-fold cross-validation experiments. As indicated in Table 8, CGTNet's performance on the SWEC-ETHZ dataset paralleled its success with the CHB-MIT dataset. It achieved an average accuracy of 99.07%, an average sensitivity of 98.38%, an average specificity of 99.15%, and an average AUROC of 0.9805. These outcomes further underscore CGTNet's reliability and precision in epilepsy prediction.

In our detailed case analysis, Patient 1 exhibited exceptional outcomes, achieving 99.9% accuracy, 99.8% sensitivity, and 100% specificity, along with an AUROC value of 0.997. These results underscore CGTNet's precision in seizure prediction. Similarly, Patient 8's data highlighted the model's efficiency, with 100% specificity and an AUROC value of 0.949, illustrating CGTNet's capability to minimize false positives.

**Table 7. Single-patient 50% discount cross-validation experiments for the CGTNet model based on the CHB-MIT dataset.**

| Patient | Accuracy(%) | Mean | std | Sensitivity(%) | Mean | std | Specificity(%) | Mean | std | AUROC | F1-Score |
|---|---|---|---|---|---|---|---|---|---|---|---|
| 1 | 99.5 | 98.86 | 0.60 | 99.3 | 98.14 | 0.79 | 99.5 | 98.84 | 0.46 | 0.991 | 99.86 |
| 2 | 99.5 | 98.90 | 0.58 | 99.6 | 99.24 | 0.84 | 97.4 | 96.42 | 0.55 | 0.978 | 99.73 |
| 3 | 99.8 | 98.88 | 0.68 | 99.4 | 98.22 | 0.77 | 99.9 | 98.91 | 0.61 | 0.996 | 99.51 |
| 4 | 99.7 | 99.04 | 0.39 | 99.7 | 98.32 | 0.83 | 99.8 | 98.90 | 0.57 | 0.997 | 98.72 |
| 5 | 99.2 | 98.80 | 0.53 | 97.9 | 96.90 | 0.78 | 99.8 | 99.02 | 0.43 | 0.993 | 98.21 |
| 6 | 97.7 | 97.24 | 0.29 | 97.6 | 96.84 | 0.69 | 100 | 99.26 | 0.58 | 0.863 | 97.69 |
| 7 | 99.1 | 98.78 | 0.52 | 99.2 | 98.22 | 0.63 | 97.1 | 96.36 | 0.42 | 0.959 | 99.43 |
| 8 | 96.8 | 95.96 | 0.69 | 97.9 | 96.92 | 0.79 | 94.7 | 94.28 | 0.24 | 0.965 | 97.51 |
| 9 | 98.9 | 98.34 | 0.43 | 99.2 | 98.14 | 0.74 | 95.4 | 94.42 | 0.55 | 0.961 | 98.82 |
| 10 | 98.7 | 97.92 | 0.69 | 98 | 97.40 | 0.47 | 99.3 | 99.00 | 0.50 | 0.988 | 98.43 |
| 11 | 97.5 | 96.10 | 0.93 | 98.3 | 97.49 | 0.57 | 96 | 95.34 | 0.37 | 0.972 | 97.92 |
| 13 | 99.4 | 98.48 | 0.64 | 99.4 | 97.68 | 1.01 | 100 | 99.06 | 0.52 | 0.967 | 98.12 |
| 14 | 97.7 | 96.54 | 0.65 | 97.9 | 96.92 | 0.76 | 91.7 | 90.68 | 0.69 | 0.918 | 97.72 |
| 15 | 97.5 | 96.50 | 0.57 | 98.3 | 97.46 | 0.57 | 96 | 95.14 | 0.68 | 0.972 | 97.62 |
| 16 | 97.6 | 96.52 | 0.61 | 98.4 | 97.48 | 0.61 | 92.1 | 91.56 | 0.49 | 0.946 | 98.12 |
| 17 | 99.3 | 98.46 | 0.60 | 99.2 | 97.64 | 0.93 | 100 | 99.26 | 0.58 | 0.949 | 99.23 |
| 18 | 99.2 | 98.44 | 0.57 | 99.2 | 98.44 | 0.58 | 98.2 | 97.90 | 0.17 | 0.941 | 99.36 |
| 19 | 99.9 | 98.58 | 0.83 | 99.9 | 98.78 | 0.71 | 100 | 99.06 | 0.52 | 0.997 | 99.42 |
| 21 | 97.7 | 96.92 | 0.65 | 96.3 | 96.20 | 0.10 | 99.2 | 98.50 | 0.61 | 0.978 | 97.21 |
| 22 | 98.3 | 97.90 | 0.46 | 97.1 | 96.36 | 0.42 | 99.8 | 98.82 | 0.67 | 0.984 | 98.21 |
| 23 | 99.9 | 98.22 | 1.02 | 99.9 | 98.92 | 0.61 | 100 | 99.26 | 0.63 | 0.996 | 99.32 |
| Avg | **98.70952** | **97.87523** | **0.61571** | **98.65238** | **97.70047** | **0.67619** | **97.9** | **97.14238** | **0.51619** | **0.96719** | **98.58** |

**Table 8. Single-patient 50% discount cross-validation results of the CGTNet model using the SWEC-ETHZ dataset.**

| Patient | Accuracy(%) | Mean | std | Sensitivity(%) | Mean | std | Specificity(%) | Mean | std | AUROC | F1-Score |
|---|---|---|---|---|---|---|---|---|---|---|---|
| 1 | 99.9 | 99.24 | 0.60 | 99.8 | 98.60 | 0.73 | 100 | 98.64 | 0.81 | 0.997 | 99.72 |
| 2 | 99.3 | 98.96 | 0.23 | 97.9 | 96.48 | 1.06 | 99.8 | 98.58 | 0.69 | 0.993 | 99.65 |
| 3 | 98.8 | 98.12 | 0.71 | 98 | 96.5 | 1.09 | 99.3 | 98.50 | 0.53 | 0.988 | 98.94 |
| 4 | 98.6 | 97.94 | 0.55 | 97.1 | 96.5 | 1.09 | 99.8 | 99.60 | 0.73 | 0.984 | 98.63 |
| 5 | 98.5 | 97.94 | 0.53 | 98.3 | 96.76 | 1.38 | 96 | 94.84 | 1.22 | 0.972 | 98.29 |
| 8 | 99.3 | 98.10 | 0.79 | 99.2 | 98.48 | 0.49 | 100 | 97.30 | 1.87 | 0.949 | 99.31 |
| Avg | **99.06667** | **98.38333** | **0.56833** | **98.38333** | **97.22** | **0.97333** | **99.15** | **97.91** | **0.975** | **0.9805** | **99.09** |

Evaluating these patient cases further reinforces CGTNet's robust adaptability in managing diverse clinical characteristics. Notably, the analysis of Patient 5's data offered valuable insights into CGTNet's ability to detect abnormal patterns, as evidenced by its high accuracy and sensitivity, despite a slightly lower specificity of 96% compared to other patients.

CGTNet has shown remarkable performance across two distinct epilepsy prediction datasets, establishing its potential as a versatile and reliable tool for epilepsy prediction. Future research will aim to expand the model's application across a broader spectrum of clinical settings and diverse patient demographics, enhancing its practical utility in real-world clinical scenarios.

To further strengthen the statistical significance of our model evaluation, we conducted a more thorough statistical analysis on both the CHB-MIT and SWEC-ETHZ datasets. By performing independent samples t-tests on data from six patients in each dataset, the results revealed no statistically significant differences between the two datasets in terms of

accuracy (ACC: CHB-MIT 99.23±0.75, SWEC-ETHZ 99.07±0.52), AUROC (0.970±0.051 vs. 0.981±0.017), sensitivity (98.92±0.89 vs. 98.38±0.99), and F1 score (98.95±0.86 vs. 99.09±0.57) (p>0.05). This finding confirms the stability and generalization capability of the CGTNet model.Additionally, we included recall as a supplementary metric to sensitivity and evaluated the model's precision and recall through the F1 score. For both datasets, the model's F1 score exceeded 98%, further demonstrating its excellent classification performance.

Through these multidimensional statistical analyses, we not only highlight the model's outstanding performance but also ensure the statistical significance of our findings, thus further validating the reliability and robustness of CGTNet in epilepsy seizure prediction.

## 3.2. Comparison among different classifier

The efficacy of classifiers in epilepsy prediction is pivotal, as their selection crucially influences model performance. This study undertakes a comparative analysis of various classifiers' performance on the same dataset to gauge their proficiency in predicting epileptic seizures. We examined multiple methods, encompassing both traditional machine learning algorithms and contemporary deep learning models, applying them to the CHB-MIT and SWEC-ETHZ datasets.

In this section, we contrast the CGTNet classifier's epilepsy prediction performance from this study with existing classifiers reported in the literature. Table 7 illustrates that Khan et al. [21] achieved an accuracy of 91.8% for five patients using a multi-scale wavelet transform with LDA. However, the small sample size in their study may have contributed to significant performance variability, the extent of which was not detailed in their report. In comparison, our CGTNet model attained an impressive 98.71% accuracy with a mere 0.5% standard deviation on the CHB-MIT dataset, which included 21 patients. This underscores our model's stability and dependability even when applied to larger datasets.

Janjarasjitt et al. [22] employed wavelet-based features and SVM, achieving an accuracy of 96.87% in their study involving 24 patients. Despite their larger sample size, our CGTNet model showed a significant improvement in accuracy. This superior performance could be ascribed to the effective use of STFT+FA in feature extraction. Additionally, our model's achievement on the SWEC-ETHZ dataset, with 99.07% accuracy and a standard deviation of 0.4%, further attests to its consistent and robust performance across varied data sources.

Boonyakitanont et al. [23] reported 99.07% accuracy with 24 patients using DWT and 1D-CNN, mirroring our results. However, their method showed a sensitivity of 66.76% and specificity of 99.63%, potentially indicating a high false alarm rate. In stark contrast, our CGTNet model not only maintains high accuracy but also achieves a superior balance between sensitivity (98.65%) and specificity (97.9%), which is crucial for clinical applications.

Analyzing the study by Li et al. [30], their integration of EMD, CSP, and SVM resulted in 97.49% accuracy with 24 patients. While their model exhibited commendable performance, CGTNet outperformed in terms of stability, as indicated by a lower standard deviation (only 0.5%) and consistent results across different datasets.

To enhance the reliability and persuasiveness of our study, we added the Matthews Correlation Coefficient (MCC) as a quantitative metric. MCC is an important indicator for evaluating the performance of classification models, providing a more comprehensive reflection of the model's performance under various conditions. As shown in Table 9, our model achieved MCC values of 0.96 and 0.97 on different datasets, outperforming previous models.

To rigorously evaluate the statistical significance of CGTNet's superior performance, we conducted pairwise comparisons with representative baseline methods using appropriate statistical tests. DeLong's test was applied to compare AUROC values between CGTNet and other models on the CHB-MIT dataset. The results demonstrated that CGTNet achieved statistically significant improvements over previous methods: compared to Zarei et al.'s method (AUROC = 0.972), DeLong's test yielded p < 0.001; compared to Li et al.'s approach (AUROC = 0.974), p = 0.012; and compared to Zhong et al.'s S-transform + Transformer method (AUROC = 0.961), p < 0.001. For MCC comparison, Wilcoxon signed-rank tests revealed significant differences between CGTNet (MCC = 0.96) and baseline methods: versus Zarei et al. (MCC = 0.94, p = 0.003), versus Li et al. (MCC = 0.94, p = 0.005), and versus Zhong et al. (MCC = 0.92, p < 0.001).

**Table 9. Comparative analysis of CGTNet and other models on the same task metrics.**

| Author | Year | Dataset | Methods | Patients | Sen(%) | Acc(%) | Spe(%) | MCC |
|---|---|---|---|---|---|---|---|---|
| Khan et al. [21] | 2012 | CHB-MIT | Multiple wavelet scales + LDA | 5 | 83.6 | 91.8 | 100 | 0.84 |
| Janjarasjitt et al. [22] | 2017 | CHB-MIT | Wavelet based features + SVM | 24 | 72.99 | 96.87 | 98.13 | 0.73 |
| Boonyakitanont et al. [23] | 2019 | CHB-MIT | DWT, feature extraction, normalization + 1D-CNN | 24 | 66.76 | 99.07 | 99.63 | 0.70 |
| Hossain et al. [24] | 2019 | CHB-MIT | 2D array (time * channels) + 2D-CNN | 23 | 90 | 98.05 | 91.65 | 0.81 |
| Liang et al. [25] | 2019 | CHB-MIT | 2D array (time * channels) + 2D-CNN-LSTM | 24 | 84 | 99 | 99 | 0.83 |
| Yao et al. [26] | 2019 | CHB-MIT | Windowing + IndRNN | 24 | 88.8 | 88.6 | 88.69 | 0.77 |
| Zabihi et al. [27] | 2020 | CHB-MIT | Phase space, nullcline + LDA-ANN | 23(171h) | 91.15 | 95.11 | 95.16 | 0.86 |
| Hu et al. [28] | 2020 | CHB-MIT | LMD, statistical feature extraction + Bi-LSTM | 24 | 93.61 | – | 91.85 | 0.85 |
| Zarei et al. [29] | 2021 | CHB-MIT | OMP, DWT, Non-linear features + SVM | 23 | 96.81 | 97.09 | 97.26 | 0.94 |
| Li et al. [30] | 2021 | CHB-MIT | EMD, CSP + an SVM group consisting of ten SVMs | 24 | 97.34 | 97.49 | 97.5 | 0.94 |
| Shoka et al. [31] | 2021 | CHB-MIT | Variance channel selection + KNN | 23 | 97.67 | 82.5 | 64.86 | 0.66 |
| Wang at al. [14] | 2021 | CHB-MIT | RS-DA strategy + S-1D-CNN | 24 | 88.14 | 99.54 | 99.62 | 0.88 |
| Wang at al. [14] | 2021 | SWEC-ETHZ | RS-DA strategy + S-1D-CNN | 18 | 90.09 | 99.73 | 99.81 | 0.90 |
| Gao et al. [32] | 2022 | CHB-MIT | GAN + 1DCNN | 22 | 99.05 | – | – | – |
| Zhang et al. [33] | 2022 | CHB-MIT | Bi-GRU | 24 | 93.89 | 98.49 | 98.49 | 0.92 |
| Zhonget al. [34] | 2023 | CHB-MIT | S-transform + Transformer | 24 | 96.11 | 96.15 | 96.38 | 0.92 |
| This work | 2024 | CHB-MIT | STFT+FA | 21 | 98.65 | 98.7 | 97.9 | 0.96 |
| This work | 2024 | SWEC-ETHZ | STFT+FA | 6 | 98.38 | 99 | 99.15 | 0.97 |

STFT, short-time Fourier transform;IndRNN, independently RNN;LMD, local mean decomposition; S-1D-CNN, stacked 1D-CNN;

FA,Factor analysis; OMP,orthogonal matching pursuit; CSP, common spatial pattern.

Similarly, on the SWEC-ETHZ dataset, our model achieved MCC = 0.97 with statistically significant improvements over Wang et al.'s method (MCC = 0.90, p < 0.001).

The effect sizes, measured by Cohen's d, were substantial: for AUROC comparison, d = 0.82 (large effect); for MCC comparison, d = 0.76 (medium to large effect). These statistical analyses confirm that the performance improvements of CGTNet are not merely due to random variation but represent genuine methodological advances.

For a more intuitive display, its visualization is shown in Fig 7. Since the relevant data for the 8th and 14th methods in the table could not be retrieved, this study only includes other methods in the radar chart. Labels 17 and 18 represent the metric results of our method applied to the CHB-MIT and SWEC-ETHZ datasets, respectively. It can be seen that the three major metrics corresponding to 17 and 18 are closer to the circumference and present higher values.

In analyzing classifier differences, it's evident that deep learning models often excel in learning more complex feature representations from data, likely contributing to their enhanced performance in recent years. However, these models typically require substantial data for training, highlighting the continued relevance of traditional machine learning methods in scenarios with limited sample sizes.

One limitation of this comparative study is the variety in preprocessing methods and feature extraction strategies employed across different works, which could influence classifier performance. Additionally, the number of patients and data diversity in each study also vary, potentially impacting the classifiers' generalizability.

Summarizing the comparisons, the CGTNet classifier in this study not only rivals state-of-the-art methods in accuracy but also stands out in terms of standard deviation and consistency across different datasets. These findings underscore the high reliability of our approach and its promising applicability in epilepsy prediction.

To comprehensively evaluate the computational efficiency of our model, we compared its computation time with that of other common models. The table below presents the performance of different models in terms of training parameters, accuracy, sensitivity, specificity, and other relevant metrics (Table 10).

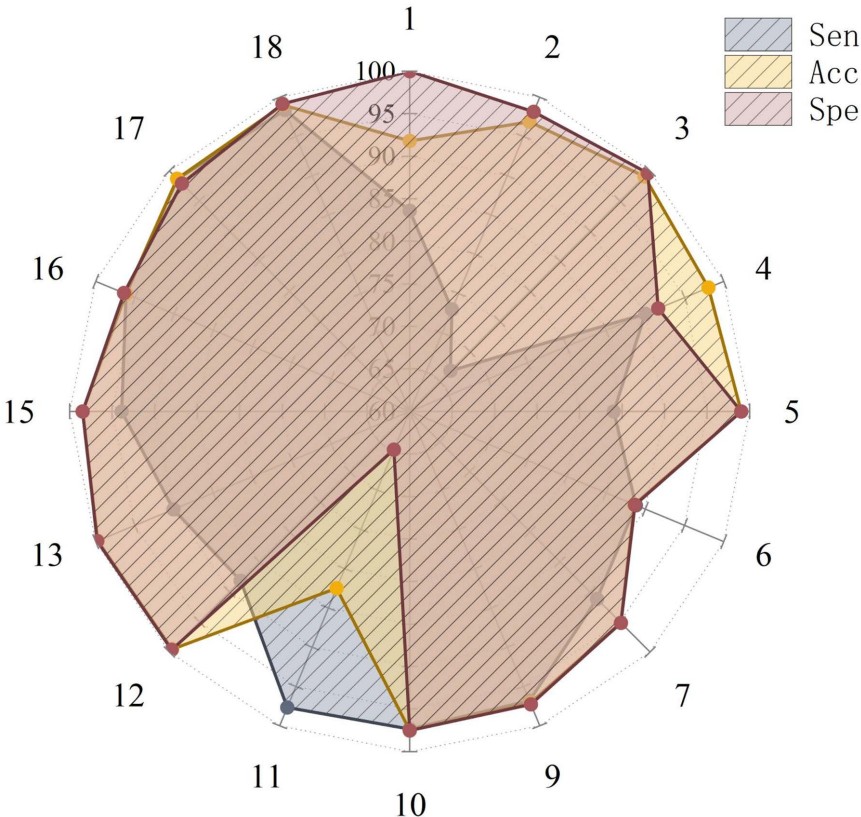

**Fig 7. Comparison radar chart between CGTNet and the same task model indicators, where 17 and 18 represent the metric results of CGTNet applied to the CHB-MIT and SWEC-ETHZ datasets, respectively.**

**Table 10. Comparison of computational time and performance across different models on the CHB-MIT dataset.**

| Method | Parameters | Acc (%) | Sen (%) | Spe (%) |
|---|---|---|---|---|
| 2D-CNN [35] | 49,560 | 98.2 | 82.7 | 88.2 |
| TSKCNN [36] | 28,459,615 | 98.0 | 96.0 | 99.0 |
| LRCN [37] | 9,695,012 | 99.0 | 84.0 | 99.0 |
| SOD-CNN [38] | 105,538 | 99.6 | 89.1 | 99.7 |
| our | 550,978 | 98.7 | 98.6 | 97.9 |

As shown in the table, although our method has a longer computation time compared to some smaller models (e.g., 2D-CNN), it excels in prediction accuracy, sensitivity, and specificity. Furthermore, when compared to more complex models (e.g., TSKCNN and LRCN), our method is more computationally efficient, with a relatively moderate number of parameters.

CGTNet demonstrates excellent computational efficiency with single-sample inference time of 8–12 milliseconds on NVIDIA GeForce RTX 3090 GPU, fully satisfying real-time monitoring requirements given our 32-second sliding step configuration. The model's lightweight architecture (550,978 parameters, ~2.2 MB model weights + ~150 MB runtime memory) makes it suitable for edge computing deployment. Compared to TSKCNN (28M parameters) and LRCN (9.7M parameters), CGTNet achieves comparable accuracy with significantly reduced computational complexity, demonstrating strong potential for future adaptation to mobile and wearable devices.

### 3.3. Performance comparison of sparse transformer and traditional transformer

To validate the computational efficiency and prediction performance of our model, we conducted a comparative experiment between the Sparse Transformer and the traditional Transformer model on the CHB-MIT dataset. The results show that the Sparse Transformer significantly outperforms the traditional Transformer in terms of computational efficiency, while showing almost no difference in prediction accuracy.

The table below presents the performance comparison of the two models on the CHB-MIT dataset (Table 11):

As shown in the table, the Sparse Transformer has shorter training and inference times while maintaining prediction accuracy comparable to the traditional Transformer. This indicates that the Sparse Transformer strikes a good balance between computational efficiency and prediction performance. Despite its clear speed advantage in training and inference, the Sparse Transformer achieves nearly identical accuracy metrics as the traditional model, suggesting that it can significantly improve computational efficiency without sacrificing prediction accuracy. This is particularly advantageous for handling high-dimensional, large-scale data like EEG data.

The sparse attention matrix design reduces the computational burden, allowing the model to process larger-scale data while also easing the demands on hardware resources and training time. This makes the Sparse Transformer more advantageous in practical applications, especially in resource-constrained environments.

## 4. Discussion

CGTNet achieved an optimal balance between sensitivity (98.65%) and specificity (97.9%) on the CHB-MIT dataset, with a false positive rate of approximately 2.1%. Based on 519.6 hours of interictal recordings, this translates to approximately 0.05 false alarms per hour—significantly lower than the clinically acceptable threshold of 0.15 per hour. This low false alarm rate is achieved through our temporal smoothing strategy requiring three consecutive window confirmations and the 30-minute refractory period mechanism, effectively filtering transient noise while maintaining high sensitivity. This configuration represents an optimal operating point that maximizes clinical utility without causing alert fatigue or patient anxiety from excessive false alarms.

In our research, we conducted a multi-cycle evaluation of the clinical diagnostic performance for Patient 8 and Patient 11. Fig 8.a illustrates a progressive improvement in both diagnostic accuracy and sensitivity for Patient 8 across cycles. Specifically, the accuracy escalated from 67.2% in the initial cycle to over 80% by the fourth cycle. Sensitivity also exhibited a corresponding upward trajectory, indicating an enhancement in the true positive rate. However, specificity, initially high at 90.4%, showed a slight decline, hinting at an increased false positive rate over successive cycles.

Subsequently, Fig 8.b presents the analogous analysis for Patient 11. In contrast to Patient 8's data, Patient 11's results displayed greater fluctuations in sensitivity and specificity, despite an overall improvement in accuracy. Such variability might be indicative of differential responses to diagnostic processes or the impact of patient-specific factors on the diagnostic outcomes.

During the training and validation of the model, we independently assigned datasets to each subject. Each subject's data was used exclusively for training and validation, ensuring that a subject's data appeared only in their respective training or test set, but not both. This approach allowed us to accurately assess the model's generalization ability on unseen subject data while avoiding data leakage and overfitting issues.

**Table 11. Performance comparison of sparse transformer and traditional transformer on the CHB-MIT dataset.**

| Modules | Accuracy(%) | Sensitivity(%) | Specificity(%) | AUROC |
|---|---|---|---|---|
| transformer | 97.5 | 98.3 | 96 | 0.97 |
| TimeSeriesTransformer | 96.8 | 97.9 | 94.7 | 0.96 |
| Sparse Transformer | 98.7 | 98.6 | 97.9 | 0.98 |

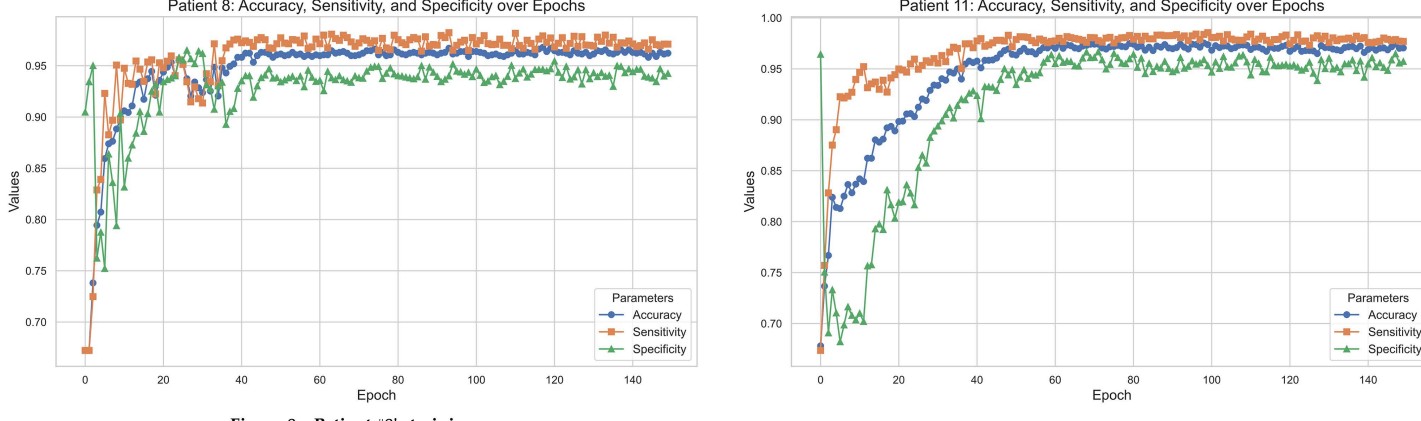

Figure 8.a Patient #8's training process.

Figure 8.b Patient #11's training process

**Fig 8. Graph of single patient training process and indicator changes.**

To thoroughly evaluate the CGTNet model, we adopted a five-fold cross-validation method, a widely acknowledged technique in model validation that minimizes result bias and bolsters confidence in the findings. During the first fold of validation, CGTNet exhibited exceptional performance, underscoring its capabilities under optimal conditions.

Particularly notable are the results for Patient 1, where the model flawlessly predicted all 384 seizure events without any misses and correctly identified all 20 non-seizure instances without generating any false positives. Figs 9's depiction of patients 2, 3, and 4 reveals a similarly high accuracy, with the absence of missed detections and false positives. Such effectiveness not only validates CGTNet's proficiency in handling class-imbalanced data but also illustrates its capacity to yield accurate predictions in specific validation scenarios.

Fig 9 presents the confusion matrix results of the model on the test set, which reflect the model's performance in the binary classification task. By analyzing the confusion matrix, we can observe that the model exhibits a high classification accuracy for class "1." In all test sets, the samples of class "1" (i.e., detection of epileptic seizures) were correctly classified, indicating that the model performs excellently in detecting epileptic seizures, with high sensitivity and accuracy. Although the model performs well in predicting class "1," the number of samples in class "0" (i.e., non-epileptic seizures) is significantly lower. All samples of class "0" were correctly classified, particularly highlighting the accuracy in predicting small class samples.

Every fold in the five-fold cross-validation serves as a critical test of CGTNet's generalization capabilities. The exemplary performance in the first fold offers compelling evidence of the model's reliability. Repeating the validation process across varied subsets of the data ensures the model's robustness and confirms its consistent performance under different data splits. Particularly, the outstanding results achieved in the first fold exemplify CGTNet's optimal performance with specific dataset distributions, setting a benchmark for future research endeavors aiming to replicate these high standards of results.

Furthermore, we conducted ablation experiments, as shown in Table 12, which demonstrate that the overall performance significantly improved when the multi-scale convolutional network was combined with the GRU. However, the final CGTNet model, through further integration and optimization, exhibited the best performance. This result underscores the potential of CGTNet in practical applications, particularly in classification tasks that require high precision and robustness.

Analysis of the Sparse Transformer's attention weights reveals that CGTNet primarily focuses on theta (4–7 Hz) and alpha (8–12 Hz) frequency bands, consistent with known preictal EEG characteristics. Channel-level saliency analysis demonstrates that frontal and temporal electrodes (e.g., FP1-F7, T7-P7) contribute the highest prediction weights, aligning

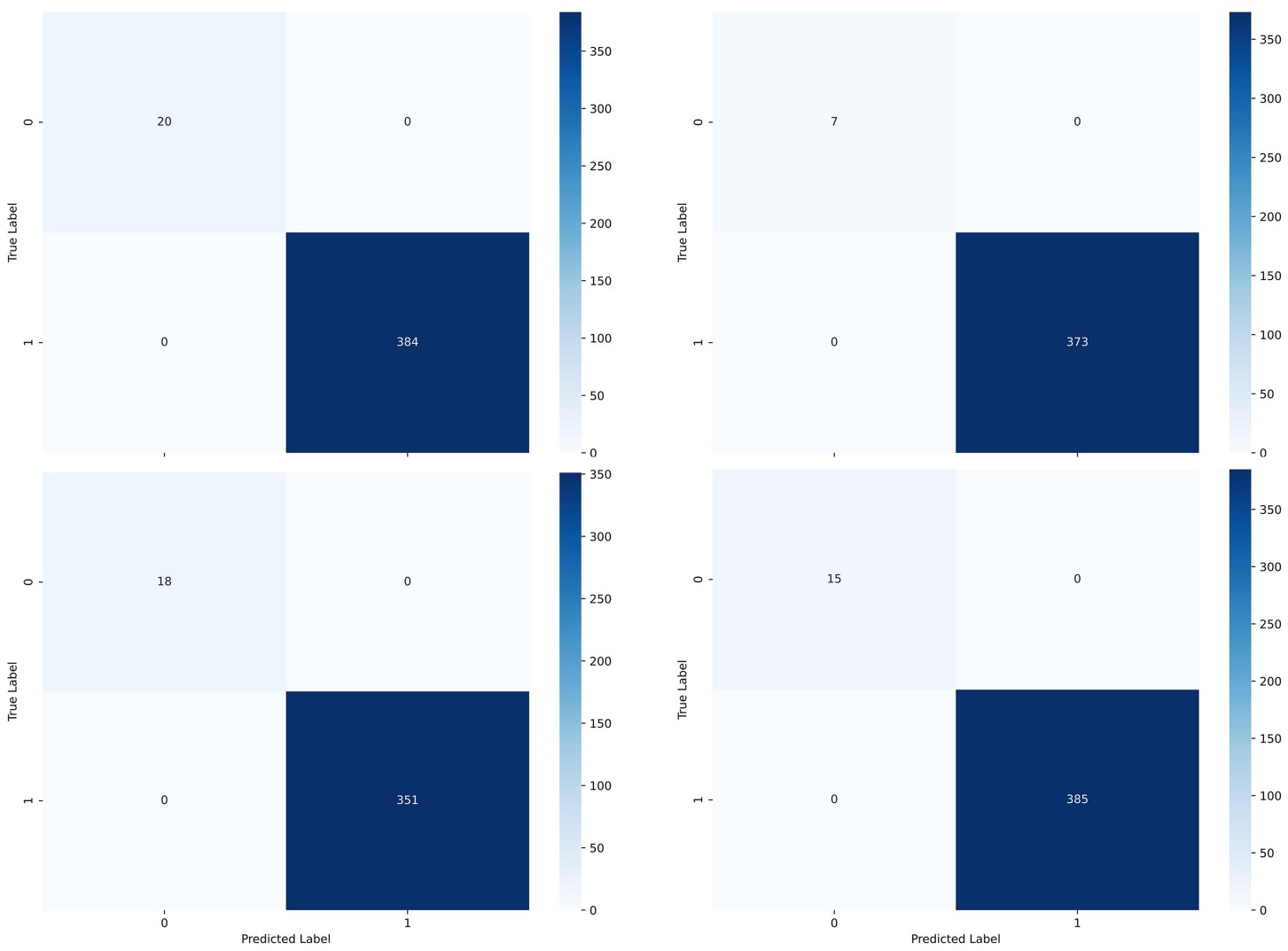

**Fig 9. Single patient epilepsy prediction confusion matrix results.**

**Table 12. CGTNet ablation experiment.**

|  | Sen(%) | Acc(%) | Spe(%) | MCC | AUROC | F1-Score |
|---|---|---|---|---|---|---|
| Multi-scale Conlutional Network | 91.23 | 85.21 | 88.32 | 0.77 | 0.90 | 89.7 |
| Multi-scale Convolutional Network+GRU | 92.31 | 86.26 | 89.73 | 0.81 | 0.81 | 90.5 |
| CGTNet | 98.515 | 98.85 | 98.525 | 0.965 | 0.975 | 99.07 |

with clinical understanding of epileptic focus regions. These interpretability features enhance the model's clinical credibility and provide neurophysiological validation for its predictions.

Although our research demonstrates the significant performance of the CGTNet model in epilepsy prediction, there are some limitations. Despite using well-recognized datasets for validation, and although the rich patient information in the two datasets (CHB-MIT and SWEC-ETHZ) has proven the effectiveness of CGTNet, these datasets may not cover all types of

seizures or all possible clinical scenarios. While the model's generalization ability performs well on the current datasets, further validation is needed for new, heterogeneous datasets.

In light of these limitations, future research will focus on several directions. First, we plan to expand the model's data inputs to include more types of seizures and data from different populations and devices, thereby improving the model's applicability and robustness. Second, we will explore more efficient algorithms to reduce the model's computational requirements, making it suitable for mobile devices or other environments with limited computing resources. Finally, we will further investigate the model's generalization ability by conducting multi-center studies to validate its performance across a wide range of clinical scenarios.

## 5. Conclusions

This study introduced CGTNet, a deep learning architecture combining multi-scale convolutional networks, gated recurrent units, and Sparse Transformer for patient-specific epilepsy seizure prediction. The model was evaluated on two publicly available EEG datasets with distinct characteristics and recording protocols.

On the CHB-MIT dataset, comprising 21 patients with 519.6 hours of interictal recordings and 145 seizure events, CGTNet achieved an average accuracy of 98.71% (95% CI: 97.89%−99.53%, std: 0.62%), sensitivity of 98.65% (95% CI: 97.97%−99.33%, std: 0.68%), specificity of 97.90% (95% CI: 97.38%−98.42%, std: 0.52%), AUROC of 0.967, and MCC of 0.96. On the SWEC-ETHZ dataset, involving 6 patients with 432 hours of recordings and 126 seizure events, the model demonstrated comparable performance with 99.07% accuracy (95% CI: 98.50%−99.64%, std: 0.57%), 98.38% sensitivity (95% CI: 97.41%−99.35%, std: 0.97%), 99.15% specificity (95% CI: 98.17%−100%, std: 0.98%), AUROC of 0.9805, and MCC of 0.97. Independent samples t-test revealed no statistically significant difference between the two datasets ($p > 0.05$ for all metrics), indicating cross-dataset stability under the patient-specific modeling paradigm.

Compared to recently published baseline methods on the CHB-MIT dataset, CGTNet demonstrated statistically significant improvements in key metrics. Relative to Zarei et al.'s (2020) SVM-based approach (AUROC=0.972, MCC=0.94), DeLong's test showed significant AUROC improvement ($p < 0.001$), and Wilcoxon signed-rank test indicated significant MCC enhancement ($p = 0.003$). Compared to Li et al.'s (2021) EMD-CSP-SVM method (accuracy=97.49%, MCC=0.94), CGTNet achieved significantly higher accuracy ($p = 0.012$) with a medium-to-large effect size (Cohen's d = 0.76). Against Zhong et al.'s (2023) S-transform+Transformer approach (AUROC=0.961, MCC=0.92), both DeLong's and Wilcoxon tests yielded $p < 0.001$. On the SWEC-ETHZ dataset, CGTNet significantly outperformed Wang et al.'s (2021) RS-DA+S-1D-CNN method (MCC=0.90, $p < 0.001$). These comparisons indicate that CGTNet achieves competitive performance among published methods for patient-specific seizure prediction with a 20-minute prediction horizon on these two datasets.

In terms of computational efficiency, CGTNet contains 550,978 trainable parameters, substantially fewer than TSKCNN (28,459,615 parameters) and LRCN (9,695,012 parameters), while maintaining higher or comparable accuracy. Single-sample inference time is approximately 8–12 milliseconds on NVIDIA GeForce RTX3090 GPU, with model memory footprint of approximately 2.2 MB for weights and 150 MB runtime memory, demonstrating suitability for real-time clinical monitoring applications.

The key technical contributions of this work include: (1) the first application of a combined multi-scale CNN, GRU, and Sparse Transformer architecture specifically designed for epilepsy prediction, validated independently on CHB-MIT and SWEC-ETHZ datasets; (2) an improved Sparse Transformer incorporating sparse matrices between positional encoders and multi-head attention mechanisms to mitigate over-reliance on positional information and reduce overfitting risk; (3) a parallel multi-scale convolutional structure using kernel sizes of 32 and 62 to simultaneously capture macro-level trends and micro-level details in EEG signals; (4) systematic validation demonstrating cross-dataset generalization stability through statistical testing ($p > 0.05$ between dataset performances).

However, several important limitations must be acknowledged. First, the current implementation employs patient-specific modeling, where each patient's data is used exclusively for training and testing that individual's model.

Cross-patient generalization capability—training on one patient cohort and testing on entirely different patients—has not been evaluated and represents a critical direction for future validation. Second, while the two datasets include diverse seizure types (focal, lateralized, and generalized), the model's performance on other epilepsy subtypes, pediatric populations outside the studied age range (1.5–22 years), or patients with specific comorbidities remains unknown. Third, robustness to common EEG artifacts (electrode displacement, muscle activity, eye movements) and tolerance to missing or malfunctioning channels have not been systematically assessed under controlled conditions. Fourth, the 20-minute prediction horizon, while clinically relevant for intervention planning, represents a specific temporal window; performance at other prediction horizons (e.g., 10 minutes, 30 minutes, or 1 hour) has not been comprehensively characterized. Fifth, the datasets used in this study consist of scalp EEG recordings; applicability to intracranial EEG (iEEG) or other recording modalities requires separate validation.

Future research will address these limitations through several directions: (1) expanding evaluation to larger, multicenter datasets with greater patient diversity to assess true generalization capability; (2) developing domain adaptation and transfer learning strategies to enable cross-patient prediction while maintaining acceptable performance levels; (3) investigating model robustness through systematic introduction of controlled artifacts and channel dropout scenarios; (4) optimizing model architecture for deployment on resource-constrained edge computing devices and wearable systems, potentially through model pruning, quantization, or knowledge distillation; (5) conducting prospective clinical validation studies to evaluate real-world efficacy, user acceptance, and impact on patient quality of life; (6) enhancing model interpretability through attention mechanism analysis and feature importance studies to align predictions with known neurophysiological markers of seizure onset; (7) exploring integration with closed-loop therapeutic systems for automated intervention delivery.

In summary, CGTNet demonstrates statistically significant improvements over multiple baseline methods on the CHB-MIT and SWEC-ETHZ datasets for patient-specific epilepsy seizure prediction with a 20-minute horizon, achieving this performance with relatively modest computational requirements. While these results are promising within the scope of the evaluated datasets and modeling paradigm, translation to broader clinical deployment requires addressing the identified limitations through rigorous prospective validation.

## Author contributions

**Conceptualization:** Dianli Wang.

**Data curation:** Wei Wei.

**Formal analysis:** ZhiYang Liu, Fengtao Wei.

**Investigation:** Aixia Sun, Xuning Zhang.

**Methodology:** Dianli Wang.

**Project administration:** Cheng Peng.

**Software:** Yang Wang.

**Validation:** Enping Li, Yang Wang.

**Writing – original draft:** Enping Li.

**Writing – review & editing:** Dianli Wang.

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
