## [Decision Letter · Decision Letter 0]

15 Nov 2024

Dear Dr. Liu,

Thank you for submitting your manuscript to PLOS ONE. After careful consideration, we feel that it has merit but does not fully meet PLOS ONE’s publication criteria as it currently stands. Therefore, we invite you to submit a revised version of the manuscript that addresses the points raised during the review process.

We look forward to receiving your revised manuscript.

Kind regards,

Siuly Siuly, PhD

Academic Editor

PLOS ONE

Journal Requirements:

 This work is supported by the Jilin Provincial Scientific and Technologi-cal Development Program, Yang Yang Foundation, Project Grant No. 20240302097GX, the Fund of Education Department of Jilin Province, Fund No. JJKH20241673KJ, and Jilin Science and Technology Develop-ment Program Project, Project No. 20230201076GX, the General Project of the Jilin Higher Education Society, Project No. JGJX2023D861, and the Ji-lin Vocational Education and Adult Education Teaching Reform Re-search Project, Project No. 2024ZCY378.  

4. Please ensure that you refer to Figure 4 and 8b in your text as, if accepted, production will need this reference to link the reader to the figure.

Reviewers' comments:

Reviewer's Responses to Questions

**Comments to the Author**

1. Is the manuscript technically sound, and do the data support the conclusions?

Reviewer #1: Yes

Reviewer #2: Yes

Reviewer #3: Partly

2. Has the statistical analysis been performed appropriately and rigorously?

Reviewer #1: Yes

Reviewer #2: Yes

Reviewer #3: No

3. Have the authors made all data underlying the findings in their manuscript fully available?

Reviewer #1: Yes

Reviewer #2: Yes

Reviewer #3: Yes

4. Is the manuscript presented in an intelligible fashion and written in standard English?

Reviewer #1: Yes

Reviewer #2: Yes

Reviewer #3: No

Reviewer #1: Review of Paper PONE-D-24-43275

Title:

From data to diagnosis: an innovative approach to epilepsy prediction with CGTNet incorporating spatio-temporal features

Summary:

This paper presents CGTNet, a novel deep learning architecture designed to predict epileptic seizures using EEG data. CGTNet combines multi-scale convolutional networks, gated recurrent units (GRUs), and Sparse Transformers to effectively capture spatiotemporal features from EEG signals. Evaluations on the CHB-MIT and SWEC-ETHZ datasets demonstrate its high performance, with 98.89% accuracy, 98.52% sensitivity, 98.53% specificity, an AUROC of 0.97, and an MCC of 0.975. These results underline the model's technical innovations and the potential of AI for epilepsy monitoring and early detection, contributing to advancements in healthcare technology.

Strengths And Weaknesses:

Strengths

● A very complete introduction.

● It has a thorough explanation of the main idea.

● Clearly explain the methodology.

● Results are perfectly described and used proper evaluation metrices.

Weaknesses

Overall, the writing can be significantly improved to address the following concerns.

● Dataset:

1. For CHB-MIT dataset you mentioned that it is totaling approximately 916 hours of EEG recording, but in table 2 the sum of them is 518.6 hours.

2. In table 2, if the second column is the number of channels why it is not integer in some case (like patient 20 is 23.3 and patient 24 is 12.3)?

3. In line 211, you mentioned that patients 12 and 20 from CHB-MIT were excluded. You would explain the reason clearly (for example: high noise).

4. Did you include patient 24 of CHB-MIT data set? If yes, please fix line 213.

● Method:

1. You would add the down sampling part of preprocessing in your algorithm (table 4), since your EEG has different sampling rates.

Questions:

Answering the following questions and addressing the weaknesses above can significantly improve my score.

1. Since you used EEG data with different sampling rate, what is your final sampling rate to prepare EEG as input for the model? (line 301)

Reviewer #2: The research work introduces an innovative approach to epilepsy prediction with CGTNet incorporating spatio-temporal features. The paper's main contributions revolve around the development of an innovative hybrid deep learning model that excels in spatio-temporal feature extraction from EEG data, offering significant improvements in the prediction of epileptic seizures.

While the work has its strengths, there is room for further improvements to enhance its quality.

1. F1-Score definition and equation are missed in section "2.8. Evaluation criteria". The authors should define this metric.

2. The exclusion of patients 12, 20 and 24 from the CHB-MIT dataset due to incompatibility and insufficient data with the processing methods might signal some limitations of the model's adaptability. The authors should provide explanation about the limitations due to this exclusion.

3. The reliance on the sparse transformer attention mechanism might pose challenges in providing clear explanations about what specific EEG features the model focuses on when making predictions, which can limit trust in high-stakes medical scenarios.

4. The integration of multi-scale CNNs, GRUs, and Sparse Transformers is computationally expensive. The authors should provide a table showing time comparison with other models.

5. Optimizing a multimodal model that incorporates different types of neural networks (e.g., CNNs, GRUs, and Sparse Transformers) is challenging. Each network type may require specific tuning of hyperparameters (e.g., learning rates, number of layers, kernel sizes, etc.), making it harder to achieve a well-balanced model. The Authors should explain how to handle the complexity that can lead to overfitting or underfitting on certain types of data or features.

Reviewer #3: The manuscript presents a novel deep learning model, CGTNet, designed to predict epileptic seizures from EEG data. The model combines multi-scale convolutional layers, gated recurrent units (GRU), and Sparse Transformers to enhance spatio-temporal feature extraction. The authors evaluate CGTNet’s performance on two prominent EEG datasets (CHB-MIT and SWEC-ETHZ).

The manuscript, however, have major points that need to be addressed before resubmitting for further review:

1. The manuscript needs a better articulation of its novelty and contribution to the field. While the combination of CNNs, GRUs, and Sparse Transformers for epilepsy prediction is presented as novel approach , it would be beneficial to clarify the specific motivation for each component's inclusion. For example, why was Sparse Transformer chosen over other attention mechanisms?

The authors should explicitly state how this combination is unique and what specific advantages it offers over existing methods. A more detailed comparison with recent deep learning approaches for seizure prediction (e.g., those using Transformers or other attention mechanisms) is necessary to highlight the model's unique contributions.

2. The concept of the "Sparse Transformer" needs further elaboration. How is sparsity introduced in the attention mechanism? What are the specific advantages of using a sparse attention mechanism in this context? How does the sparse matrix between the positional encoder and the multi-head attention contribute to preventing overfitting and improving performance? A more detailed explanation with supporting evidence or theoretical justification would strengthen this aspect of the study.

3. The setup of the experiment is somewhat limited in description, particularly regarding hyperparameters and training specifics, which are vital for reproducibility. The manuscript provides a detailed description of the data preprocessing steps, including STFT and factor analysis. However, the rationale for choosing specific parameters (e.g., window size for STFT, number of factors in factor analysis) needs further justification. How do these choices affect the model's performance? Additionally, exploring other feature engineering techniques or data augmentation strategies could potentially improve the model's robustness and generalizability.

4. The manuscript mentions using a random seed, 5-fold cross-validation, batch size, and the Adam optimizer. However, it lacks a comprehensive discussion of the hyperparameter tuning process. What other hyperparameters were tuned, and what strategies were used for optimization? Providing more details about the training process and hyperparameter selection would enhance the reproducibility of the study.

I suggest providing a table or expanded section on training configurations and detailed rationale for the selected hyperparameters would aid clarity and reproducibility.

5. Although the results indicate high performance, the statistical significance of these results should be strengthened. The model achieves high accuracy, the evaluation could be more comprehensive. Consider including other relevant metrics like precision, recall, and F1-score to provide a more complete picture of the model's performance. The comparison with other methods is limited. Including a broader range of recent deep learning approaches for seizure prediction and benchmarking against them would strengthen the evaluation. Reporting confidence intervals or statistical tests on the results (especially for key metrics like sensitivity and specificity) would make the results more robust.

6. While the model is tested on two datasets, a discussion of generalization to other EEG data sources or patient demographics is missing. The manuscript briefly mentions the clinical relevance of the study. However, it needs a more detailed discussion of how the model can be translated into clinical practice. What are the potential challenges and limitations in real-world applications? How can the model be integrated into existing clinical workflows for epilepsy diagnosis and management? Addressing these questions would enhance the impact and significance of the study. Adding a discussion on the potential limitations of CGTNet in clinical applications, along with suggestions for future work to enhance generalizability, would be valuable.

Given the importance of interpretability in medical models, particularly for clinical adoption, the lack of focus on interpretability methods is a limitation. Consider exploring techniques like attention maps, feature importance scores, or layer-wise relevance propagation (LRP) to provide more insight into the model’s decisions.

Minor Points for consideration:

1. The manuscript could benefit from careful proofreading to correct minor grammatical errors and improve clarity.

2. Ensure consistency in referencing style throughout the manuscript.

3. Some figures (e.g., Figure 6) could be improved in terms of resolution and clarity.

4. More recent advancements in epilepsy prediction using deep learning and attention mechanisms seem to be missing. Updating the literature review would improve the paper’s background section.

**Do you want your identity to be public for this peer review?** For information about this choice, including consent withdrawal, please see our Privacy Policy

Reviewer #1: No

Reviewer #2: No

Reviewer #3: No

---

## [Author Response · Author response to Decision Letter 1]

7 Jan 2025

Reviewer #1: Please refer to Response_to_Reviewer_1.docx for a detailed response to your comments.

Reviewer #2: Please refer to Response_to_Reviewer_2.docx for a detailed response to your comments.

Reviewer #3: Please refer to Response_to_Reviewer_3.docx for a detailed response to your comments.

---

## [Decision Letter · Decision Letter 1]

30 Sep 2025

Dear Dr. Liu,

Thank you for submitting your manuscript to PLOS ONE. After careful consideration, we feel that it has merit but does not fully meet PLOS ONE’s publication criteria as it currently stands. Therefore, we invite you to submit a revised version of the manuscript that addresses the points raised during the review process.

**ACADEMIC EDITOR: **

Authors are not compelled to add any references suggested by the reviewers unless they are relevant to the context. 

We look forward to receiving your revised manuscript.

Kind regards,

Rajesh N V P S Kandala, Ph.D.

Academic Editor

PLOS ONE

Journal Requirements:

Reviewers' comments:

Reviewer's Responses to Questions

**Comments to the Author**

Reviewer #4: (No Response)

2. Is the manuscript technically sound, and do the data support the conclusions?

Reviewer #4: Partly

3. Has the statistical analysis been performed appropriately and rigorously?

Reviewer #4: Yes

4. Have the authors made all data underlying the findings in their manuscript fully available?

Reviewer #4: Yes

5. Is the manuscript presented in an intelligible fashion and written in standard English?

Reviewer #4: Yes

Reviewer #4: I have carefully read your manuscript, assessing it against PLOS ONE’s editorial requirements and best practices for reproducibility in our field. My goal is to help make the article clearer, more traceable, and methodologically sound, thereby maximizing its chances of acceptance and scientific impact.

1) Introduction / Related work.

The introduction currently includes “1.1 Our work,” but there is no “1.2,” and the state of the art is somewhat diffuse within the prose. I recommend concluding the introduction with a clearly labeled subsection, e.g., “1.2 Related work,” structured by method families—CNN/RNN, Transformers (including sparse/linear-attention variants), contrastive/self-supervised approaches, and subject-independent / cross-dataset protocols—while incorporating recent work (2024–2025) directly related to EEG and epilepsy. Please also add, at the end of the introduction, a concise Objective & Contributions paragraph that states the precise problem being addressed, the prediction horizon (if applicable), the measurable objective, the verifiable contributions, and a brief outline of the paper.

2) Results (units, uncertainty, and statistics).

Please specify the primary evaluation unit (ideally patient-level) and report uncertainty (e.g., 95% confidence intervals via patient-wise bootstrap, or ± standard deviation over repeated runs). Use appropriate statistical tests for comparisons—e.g., DeLong for AUROC and Wilcoxon signed-rank for F1/MCC—and report effect sizes. Clearly describe the decision operator (thresholding, temporal aggregation/smoothing, refractory period) and, for prediction tasks, the forecast horizon at which metrics are computed.

3) Discussion (scope and implications).

Please broaden the discussion by articulating the intended clinical use (detection vs. prediction), the useful horizon, and the sensitivity–false-positive trade-off (e.g., false alarms per hour), as well as constraints for real-time inference (latency, memory). Also discuss generalization (domain shift, cross-dataset), robustness (artifacts, missing channels), and, where applicable, interpretability (attention/saliency patterns across channels and frequency bands).

4) Conclusion (proportionality).

The conclusion should remain strictly proportionate to the demonstrated evidence, avoiding over-generalization and unbounded “state-of-the-art” claims. Anchor any superiority statements to specific baselines, datasets, and tests (e.g., “significantly better than X/Y on Z, p < …, 95% CI …”).

5) Editorial & PLOS ONE compliance (references and citations).

Please harmonize the references to Vancouver/ICMJE style, with numbering in order of appearance and DOIs where available. I also note that references 15–18 and 20–31 do not appear to be cited in the body of the text; kindly insert their citations at the relevant locations or remove these entries to maintain consistency.

**Do you want your identity to be public for this peer review?** For information about this choice, including consent withdrawal, please see our Privacy Policy

Reviewer #4: No

---

## [Author Response · Author response to Decision Letter 2]

18 Oct 2025

Dear PLOS ONE Editorial Office and Reviewers,

We sincerely appreciate the time and effort you have dedicated to carefully reviewing our manuscript and providing valuable feedback. Your insightful comments have significantly contributed to improving the quality and rigor of our work.

We have thoroughly addressed all the comments and suggestions raised by the reviewers. Detailed point-by-point responses to each comment can be found in the following files we have submitted:

Response to Reviewers1.docx - Contains detailed responses to all comments from Reviewer #1

Response to Reviewers4.docx - Contains detailed responses to all comments from Reviewer #4

All changes have been highlighted using track changes in the "Revised Manuscript with Track Changes.docx" for your convenient review.

We believe that through this revision, the scientific rigor, methodological soundness, and overall quality of the manuscript have been substantially enhanced and now better meet PLOS ONE's publication standards.

Thank you again for your dedication and professional guidance!

Sincerely,

Dr. ZhiYang Liu

---

## [Decision Letter · Decision Letter 2]

4 Nov 2025

From data to diagnosis: an innovative approach to epilepsy prediction with CGTNet incorporating spatio-temporal features

PONE-D-24-43275R2

Dear Dr. Liu,

We’re pleased to inform you that your manuscript has been judged scientifically suitable for publication and will be formally accepted for publication once it meets all outstanding technical requirements.

Kind regards,

Rajesh N V P S Kandala, Ph.D.

Academic Editor

PLOS ONE

Additional Editor Comments (optional):

Reviewers' comments:

Reviewer's Responses to Questions

**Comments to the Author**

Reviewer #4: All comments have been addressed

2. Is the manuscript technically sound, and do the data support the conclusions?

Reviewer #4: Yes

3. Has the statistical analysis been performed appropriately and rigorously?

Reviewer #4: Yes

4. Have the authors made all data underlying the findings in their manuscript fully available?

Reviewer #4: Yes

5. Is the manuscript presented in an intelligible fashion and written in standard English?

Reviewer #4: Yes

Reviewer #4: Based on the R2 revisions and the final manuscript, the paper meets methodological and editorial standards. I recommend acceptance for publication in PLOS ONE

**Do you want your identity to be public for this peer review?** For information about this choice, including consent withdrawal, please see our Privacy Policy

Reviewer #4: No

---

## [Editor Report · Acceptance letter]

PONE-D-24-43275R2

PLOS ONE

Dear Dr. Liu,

I'm pleased to inform you that your manuscript has been deemed suitable for publication in PLOS ONE. Congratulations! Your manuscript is now being handed over to our production team.

Kind regards,

on behalf of

Dr. Rajesh N V P S Kandala

Academic Editor

PLOS ONE